# INPUT-ADAPTIVE BAYESIAN MODEL AVERAGING

## ABSTRACT

This paper studies prediction with multiple candidate models, where the goal is to combine their outputs. This task is especially challenging in heterogeneous settings, where different models may be better suited to different inputs. We propose input adaptive Bayesian Model Averaging (IA-BMA), a Bayesian method that assigns model weights conditional on the input. IA-BMA employs an input adaptive prior, and yields a posterior distribution that adapts to each prediction, which we estimate with amortized variational inference. We derive formal guarantees for its performance, relative to any single predictor selected per input. We evaluate IA-BMA across regression and classification tasks, studying data from personalized cancer treatment, credit-card fraud detection, and UCI datasets. IA-BMA consistently delivers more accurate and better-calibrated predictions than both non-adaptive baselines and existing adaptive methods.

## 1 INTRODUCTION

Many applications require *adaptive predictions*. In personalized medicine, different patients respond differently to the same treatment (Mahajan et al., 2023); in fairness-sensitive domains, predictions need to adapt to subpopulations (Wang et al., 2019; Grother et al., 2019); and in fraud detection, behavioral data is often heteroskedastic and varies substantially across inputs (Varmedja et al., 2019).

When the data is complex, selecting a single model that performs well across all inputs is challenging. This motivates *model averaging* (MA), which produces an *ensemble* of models. This idea dates back at least to the 1960s (see, e.g., (Clemen, 1989) for a historical perspective).

We denote data points by $x \in \mathcal{X}$, labels by $y \in \mathcal{Y}$, and the space of probability distributions on labels by $\mathcal{P}(\mathcal{Y})$. MA combines the predictive distributions of $m$ models $\{f_j : \mathcal{X} \to \mathcal{P}(\mathcal{Y})\}_{j=1}^m$ into a weighted ensemble, $p_\alpha(y \mid x) \coloneqq \sum_{j=1}^m \alpha_j f_j(y \mid x)$, with weights $\alpha_j > 0$ (often constrained to sum to one). MA accounts for the possibility that multiple models can provide plausible explanations of the data.

In classical MA, the same weights $\alpha_1, \ldots, \alpha_m$ are used for all inputs $x$. But in practice, different values of the input $x$ might call for different predictive models. This motivates *adaptive averaging*, where the weights $\alpha_j$ depend on $x$:

$$\alpha : \mathcal{X} \to \Delta^{m-1}, \qquad x \mapsto \alpha(x) = (\alpha_1(x), \ldots, \alpha_m(x)). \tag{1}$$

The result is an adaptive weighted prediction,

$$p_\alpha(y \mid x) \coloneqq \sum_{j=1}^m \alpha_j(x) f_j(y \mid x). \tag{2}$$

This model is also known as a *mixture of experts* (Jacobs et al., 1991; Jordan & Jacobs, 1994), where the adaptive weights $\alpha_j(x)$ are fit to maximize the predictive log likelihood of the data.

In this paper, we take a Bayesian perspective. We assume that the set of predictors $\mathcal{F} \coloneqq \{f_1, \ldots, f_m\}$ is fixed, and model the selection of a predictor as a random process. Our model constructs a *random selector* $g : \mathcal{X} \to \{e_1, \ldots, e_m\}$ where $\{e_j\}_{j=1}^m$ denote $m$ indicator vectors, i.e., $g(x) = e_j$ selects predictor $f_j$. Moreover, the prior on $g$ itself depends on the inputs $x$. Therefore, in our model, adaptivity arises not only from the variability of $g(x)$ across inputs, but also from allowing its prior to vary with $x$.

Under this model, MA is a natural consequence of the posterior predictive distribution. Consider a dataset $\mathcal{D} := \{x_i, y_i\}_{i=1}^n$. The posterior predictive distribution for a new input $x$ is

$$p(y \mid x, \mathcal{D}) = \sum_{j=1}^m f_j(y \mid x)\, p(g(x) = e_j \mid x, \mathcal{D}), \qquad (3)$$

where $p(g(x) = e_j \mid x, \mathcal{D})$ is a *data dependent posterior* that incorporates both training inputs and labels. Eq. 3 is an ensemble of candidate models, with weights $\alpha_j(x)$ equal to the posterior over $g$:

$$\alpha_j(x) = p(g(x) = e_j \mid x, \mathcal{D}). \qquad (4)$$

Unlike maximum likelihood approaches to MoE, this posterior captures the uncertainty over which predictor is most plausible for each input $x$.

Below, we first analyze the theoretical advantages of this adaptive Bayesian model averaging framework and derive finite-sample guarantees that compare its performance to that of any single predictor selected per input (Section 2.1). We then develop input adaptive Bayesian Model Averaging (IA-BMA), by (i) constructing an input adaptive prior, following Slavutsky & Blei (2025), and (ii) employing amortized variational inference to approximate the posterior (Section 3). We evaluate IA-BMA across regression and classification benchmarks (Section 4), and show that IA-BMA achieves substantial gains in both accuracy and calibration compared to existing adaptive, and non-adaptive strategies.

## 1.1 RELATED WORK

MA is regarded as the machine learning analogue of the "Condorcet's jury" theorem (Mennis, 2006), leveraging the "wisdom of the crowd" to mitigate the inherent uncertainty in model selection. Thus, MA is often used when there are alternative, potentially overlapping hypotheses and no clear justification for selecting a single preferred model. Applications include ecological research (Wintle et al., 2003; Thuiller, 2004; Richards, 2005; Dormann et al., 2008; Lauzeral et al., 2015; Zheng et al., 2024) and medicine (Jiang et al., 2021; Nanglia et al., 2022; Mahajan et al., 2023). More broadly, MA has been adopted in a wide range of machine learning tasks (e.g., Fernández-Delgado et al. (2014), Rokach (2010)).

As a form of model combination, MA is closely related to other ensemble techniques such as bagging (Breiman, 1996) and boosting (Freund, 1995). It is a variant of stacking procedure (Wolpert, 1992), in which outputs of base learners are combined to produce the final prediction.

MA has been shown to reduce prediction errors beyond those of the best individual component model (Dormann et al., 2018; Peng & Yang, 2022) and to mitigate overfitting (Dietterich et al., 2002; Polikar, 2006). In recent years, extensive surveys have reviewed MA (Kulkarni & Sinha, 2013; Woźniak et al., 2014; Gomes et al., 2017; González et al., 2020; Sagi & Rokach, 2018; Wu & Levinson, 2021), with some focusing specifically on decision trees (Rokach, 2016) or neural networks (Ganaie et al., 2022).

A Bayesian method for MA was introduced by Waterhouse et al. (1995), who place a prior directly on the averaging weights. In contrast, we reinterpret MA as a problem of random model selection, leading to *dynamic* model selection in which the choice of model adapts to the specific input. Earlier work on dynamic model selection includes Cao et al. (1995); Giacinto & Roli (1999); Gunes et al. (2003); Didaci et al. (2005); Didaci & Giacinto (2004). However, these approaches focus on selecting a single model for each instance, rather than assigning instance-specific weights to average predictions across multiple models.

**Input adaptive model averaging methods:** Few methods assign input-dependent weights. These date back to Mixture of Experts (MoE) (Jacobs et al., 1991), where a gating network maps the input $x$ to weights $\alpha_j(x)$, estimated by maxmizing the induced likelihood. Classical MoE variants jointly train both experts and gates, and an extensive literature explores different expert classes and gating architectures (see (Yuksel et al., 2012) for a review). In our setting, however, we consider the MoE variant in which the gating network is applied on top of pre-trained experts.

Rasmussen & Ghahramani (2001) extended this framework by using Gaussian Processes (GPs) as base models, providing nonparametric flexibility. They adopt a Bayesian perspective with a Dirichlet

Process (DP) prior, yielding an infinite mixture. However, here weights and base models are learned jointly, and thus only a single family of base predictors is considered.

Although these methods often outperform standard model averaging, maximum-likelihood–based assignment tends to concentrate probability mass on the predictor that is most confident about the observed outcome $y$, frequently resulting in overconfident predictions (Freund & Schapire, 1997; Guo et al., 2017). Several approaches proposed alternative strategies for weight assignment.

Woods et al. (1997) proposed a dynamic scheme based on local accuracy estimates. For a test input $x$, its neighborhood is identified (typically via $k$-nearest neighbors), and each classifier's performance in this region is summarized as a local accuracy score. The classifier with the highest score is then selected to predict $x$.

Similarly, Chan & van der Schaar (2022) proposed an approach that assigns higher weight to models whose training domains better cover a test instance. Inputs are mapped into a learned low-dimensional space where models with similar predictions are closer together, and weights are set via kernel density estimation. Unlike Woods et al. (1997), where similarity is predefined, here it is learned from data. Motivated by Tenzer et al. (2022), the method assumes that models making random errors on an input are unlikely to agree.

Perhaps most relevant to our work is Bayesian hierarchical stacking (BHS) (Yao et al., 2022), which places priors on logit weights, and models them with hierarchical low-rank linear functions. The parameters are then estimated by maximizing the expected log predictive density.

Thus, prior work on adaptive model averaging has focused predominantly on methods targeting frequentist objectives, with relatively few Bayesian formulations. In contrast to previous approaches, our model assumes a fully Bayesian setting in which the selector itself is random and, crucially, is defined locally relative to each input $x$. This yields an input-dependent prior $p(g \mid x)$ rather than a global prior $p(g)$. In turn, this prior induces an adaptive posterior that corresponds exactly to the Bayes-optimal weights, providing a principled approach for adaptive model averaging.

## 2 PROBABILISTIC FORMULATION OF ADAPTIVE MODEL AVERAGING

We cast adaptive model averaging as a probabilistic model selection. To reflect that some models may be better suited for different inputs, we assume a probabilistic model in which the *selection function* $g$ is treated as a random input-dependent variable. For a training set $\mathcal{D} = \{(x_i, y_i)\}_{i=1}^n$, and a new input $x$, we assume the data generating process

$$x_i, x \overset{\text{iid}}{\sim} p(x), \tag{5}$$

$$g \sim p(g \mid x, x_{1:n}), \tag{6}$$

$$y_i \sim p(y_i \mid x_i, g), \ y \sim p(y \mid x, g). \tag{7}$$

We defer the precise specification of the adaptive prior $p(g \mid x, x_{1:n})$ to Section 3.1.

The predictive distribution for $y$ given a new input $x$ and the training data is then

$$p(y \mid x, \mathcal{D}) = \int p(y \mid x, \mathcal{D}, g)\, p(g \mid x, \mathcal{D})\, d\mu(g) = \int p(y \mid x, g)\, p(g \mid x, \mathcal{D})\, d\mu(g), \tag{8}$$

where $p(g \mid x, \mathcal{D})$ is a posterior distribution on the space of functions[1] $\mathcal{G} := \{g : \mathcal{X} \to \{e_1, \ldots, e_m\}\}$.

A draw from the posterior $g \sim p(g \mid x, \mathcal{D})$ induces a random index $J(x)$, defined by the relation $g(x) = e_{j(x)}$. Using this index, we can rewrite equation 8 as

$$p(y \mid x, \mathcal{D}) = \int p(y \mid x, g)\, p(g \mid x, \mathcal{D})\, d\mu(g) = \sum_{j=1}^m f_j(y \mid x)\, p(J(x) = j \mid x, \mathcal{D}). \tag{9}$$

A formal proof of this equality is outlined in Appendix A.1.

---

[1] Formally, $p(g \mid x, \mathcal{D})$ is a density w.r.t some reference measure $\mu$ on a space of measurable functions $\mathcal{G}$.

Under our model, the predictive distribution is a mixture of the candidate predictions $f_j(y \mid x)$ weighted by the posterior probabilities $p(J(x) = j \mid x, \mathcal{D})$. In other words, the input adaptive weights $\alpha_j(x)$ *arise directly from the probabilistic formulation* itself, and *they are precisely the posterior probabilities* of each model being the generator at input $x$.

A central difficulty, of course, is that the true posterior is unknown. In Section 3, we introduce a variational approximation to $p(J(x) = j \mid \mathcal{D}_i, x)$ that preserves explicit dependence on both $x$ and $\mathcal{D}$. Before presenting this approximation, we first analyze the performance guarantees that arise when the averaging weights are set to the true posterior probabilities $p(J(x) = j \mid x, \mathcal{D})$.

## 2.1 LIKELIHOOD GUARANTEES

So far we have seen that the posterior probabilities $p(J(x) = j \mid x, \mathcal{D})$ arise naturally as input adaptive weights under our model. In particular, they are the Bayes-optimal weights, as they recover the true predictive distribution.

We now show that this choice also comes with performance guarantees: the posterior-weights predictor not only reflects the correct probabilistic formulation, but in expectation achieves likelihood performance competitive with any input-specific single-model selector. The next theorem formalizes this result (for proof see Appendix A.2).

**Theorem 2.1.** *Denote $\mathcal{D}_i := \{(x_t, y_t)\}_{t=1}^i$, and consider the posterior weights predictor $\hat{p}_\alpha^{(i)}$ assigning $\alpha_j(x; \mathcal{D}_i) = p(J(x) = j \mid \mathcal{D}_i, x)$ to the $j$-th predictor $f_j$. Assume that $\mathbb{E}[|\log f_j(Y \mid X)|] < \infty$ for all $f_j \in \mathcal{F}$. Then, for any measurable selector $j^* : \mathcal{X} \to \{1, \ldots, m\}$ and any $n \geq 1$,*

$$\frac{1}{n} \sum_{i=1}^n \mathbb{E}\left[\log \hat{p}_\alpha^{(i)}(y_i \mid x_i, \mathcal{D}_{i-1})\right] \geq \mathbb{E}\left[\log f_{j^*(x)}(y \mid x)\right] + \frac{1}{n} \sum_{i=1}^n \mathbb{E}\left[\log \alpha_{j^*(x_i)}^{(i)}(x_i)\right], \quad (10)$$

*where the expectations are taken w.r.t the population distribution $(x_i, y_i) \sim p(x, y)$.*

Thus, the posterior weights predictor can match any per-input selector (i.e., a rule that may pick a different $j$ for different $x$), up to a term depending on the gating weights assigned to the chosen model at each $x$. Put plainly, the posterior mean performs nearly as well as if we could select the best expert separately for every $x$.

Concretely, for the selector that picks the most probable model, $j^{(i)}(x) \in \arg\max_{1 \leq j \leq m} \alpha_j^{(i)}(x)$, the penalty becomes $\frac{1}{n} \sum_{i=1}^n \mathbb{E}\left[\log \max_j \alpha_j^{(i)}(x_i)\right]$, which vanishes as the posterior sharpens, i.e., when $\max_j \alpha_j^{(i)}(x_i) \to 1$ in probability.

# 3 IA-BMA: INPUT ADAPTIVE BAYESIAN MODEL AVERAGING

Our *goal* is to develop a method for estimating this posterior distribution over models. By doing so, we obtain an averaging scheme that is consistent with both the training data $\mathcal{D}$ and the specific input $x$, thereby approximating the true predictive distribution that we ultimately aim to recover.

We begin by formulating the modeling assumptions for an adaptive prior that is conditioned jointly on the training covariates and a new input. Building on this prior, we then develop a variational inference method to approximate the resulting posterior.

## 3.1 ADAPTIVE PRIOR

Based on the adaptive prior introduced in (Slavutsky & Blei, 2025), we posit a prior that encodes the plausibility of each model conditional on both the training covariate $x_{1:n}$ and a new input $x$ at which prediction is sought. This prior is defined through an energy-based formulation.

Specifically, for a predictor $f_j$ we consider the prior induced by the negative energy function

$$E(J = j; x_{1:n}, x) := \int \sum_{i=1}^{n} \log p(y|x_i, f_j) + \log p(y|x, f_j) \, dy \tag{11}$$

$$p(J = j|x_{1:n}, x) := \frac{1}{Z(f)} \exp\left(E(J = j; x_{1:n}, x)\right), \tag{12}$$

where the normalizing factor[2] is given by $Z(f) := \sum_{j=1}^{m} \exp\left(E(J = j; x_{1:n}, x)\right)$.

This prior allows beliefs about model plausibility to adapt to the new input $x$. Unlike a prior defined solely from the training data, which remains fixed across prediction points, our formulation updates the relative weight of each model once $x$ is observed. This makes the prior *input adaptive*, enabling model selection probabilities to shift dynamically with the prediction covariates. To build intuition, we next examine a simple analytical example.

Thus, IABMA adds a second layer of adaptivity: the prior itself varies with $x$, linking each expert's prior selection probability to its expected likelihood and propagating this uncertainty into the posterior.

In section 4 we show that our prior indeed rewards predictors whose likelihood is high locally at $x$, and quantify the additional improvement stemming from this prior in Appendix B.6.

**A two-model Bernoulli example** Suppose $y \in \{0, 1\}$, and consider two candidate logistic models

$$p(y = 1 \mid x, f_j) = \sigma(\beta_j x), \qquad \sigma(u) := \frac{1}{1 + e^{-u}}, \tag{13}$$

with $j \in 1, 2$ and slopes $0 < \beta_2 < \beta_1$. In this setting, the energy function is given by

$$E(J = j; x_{1:n}, x) = \sum_{i=1}^{n} \sum_{y \in \{0,1\}} \log p(y \mid x_i, f_j) + \sum_{y \in \{0,1\}} \log p(y \mid x, f_j) \tag{14}$$

$$= \underbrace{\sum_{i=1}^{n} \log\left(\sigma(\beta_j x_i)\left[1 - \sigma(\beta_j x_i)\right]\right)}_{=:C_j} + \underbrace{\log\left(\sigma(\beta_j x)\left[1 - \sigma(\beta_j x)\right]\right)}_{=:\ell_j(x)} \tag{15}$$

and the adaptive prior is

$$p(J = j \mid x_{1:n}, x) = \frac{\exp(C_j + \ell_j(x))}{\sum_{k=1}^{m} \exp(C_k + \ell_k(x))} \tag{16}$$

Accordingly, the log-odds between the two models is

$$\log \frac{p(J = 1 \mid x_{1:n}, x)}{p(J = 2 \mid x_{1:n}, x)} = (C_1 - C_2) + \ell_1(x) - \ell_2(x). \tag{17}$$

Thus, the log-odds depend both on the difference between training baselines $C_1 - C_2$, and the change induced by conditioning also on the new input $x$ is $\delta_x := \ell_1(x) - \ell_2(x)$.

Concretely, suppose the baseline difference is fixed at $C_1 - C_2 = \log 5 \approx 1.61$, yielding $p(J = 1 \mid \mathcal{D}) = \sigma(\log 5) \approx 0.83$. Based solely on the training data, the prior thus strongly favors $f_1$. Now consider a new input $x = 1$ with $\beta_2 = 1$. As $\beta_1$ increases, the discrepancy $|\ell_1(1) - \ell_2(1)|$ grows, and $\delta_{x=1}$ shifts the likelihood ratio toward $f_2$. For example, when $\beta_1 = 3$, the prior shifts to a mild preference for $f_1$, at $\beta_1 = 5$ it flips to favor $f_2$, and by $\beta_1 = 9$ the preference for $f_2$ becomes very strong. These dynamics, along with additional parameter settings for coefficients and baseline differences, are shown in Figure 1.

This analysis highlights the interplay between the baseline preference and the input-specific adjustment introduced by $x$. It shows that, in extreme cases, even strong baseline beliefs can be overturned by the adaptive correction at the queried input.

---

[2]This definition requires integrability of $\exp\left(E(J = j; x_{1:n}, x)\right)$, and thus we assume that $\exp(E(J = j; x_{1:n}, x))$ is integrable for each $j$.

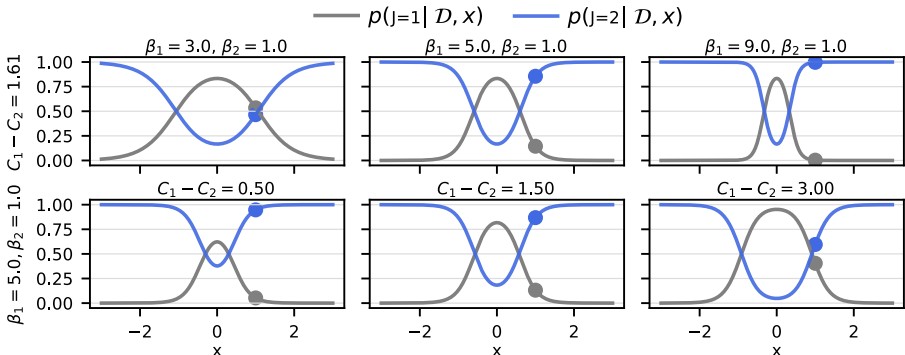

Figure 1: Illustration of the input adaptive prior. Each panel shows the posterior probabilities $p(J = j \mid \mathcal{D}, x)$ as functions of $x$. *Top*: the baseline log-odds is fixed and $\beta_1$ varies; larger $\beta_1$ values increase the influence of $x$, producing stronger adaptive corrections. *Bottom*: $\beta_1, \beta_2$ are fixed while the baseline log-odds $C_1 - C_2$ varies; stronger baselines yield higher prior preference for $f_1$, but input-specific corrections can still substantially reshape the prior at certain $x$. The marked point $(x = 1)$ highlights how the adaptive prior shifts the relative model probabilities compared to the baseline.

**Evaluation of the prior:**   Evaluating the proposed prior requires computing an integral over the outcome space $\mathcal{Y}$, and thus depends on whether the outcome space is discrete or continuous. When $\mathcal{Y}$ is *discrete* (e.g., in classification problems), the integral reduces to a finite sum over all possible outcome values. In this case, the evaluation is straightforward and can be computed exactly without approximation. When $\mathcal{Y}$ is *continuous*, (e.g., in regression problems), the integral cannot typically be computed in closed form and may even diverge unless we restrict the domain of integration. Thus, to approximate the prior, as in (Slavutsky & Blei, 2025), we employ Monte-Carlo integration where we sample $K$ possible outcome values uniformly from a predefined integration range $[y_{\min}, y_{\max}]$ set to large margin upon observed values in training data, and average the Normal log-likelihood (centered at the model's prediction with unit variance) over the $K$ samples.

This procedure introduces no meaningful computational overhead: in classification it reduces to a simple summation, and in regression we approximate the expectation using $K = 64$ Monte-Carlo samples, which we found to be numerically stable in practice. Table 18 confirms that runtime remains comparable to Mixture-of-Experts and DDP models using the same architecture.

## 3.2 Amortized Variational Posterior

Equipped with the adaptive prior, we now turn to the estimation of the posterior $p(J = j \mid x_{1:n}, y_{1:n}, x) = p(J = j \mid \mathcal{D}, x)$, which conditions not only on the covariates $x$ and $x_{1:n}$, but also on the training labels $y_{1:n}$. This, in turn, will enable us to assign input adaptive weights for model averaging, bringing them closer to the ideal weights that recover the predictive distribution $p(y \mid x)$.

We do so by fitting variational distributions $q(f_j; x) \approx p(J = j \mid \mathcal{D}, x)$ parameterized as functions of the input $x$. This yields an *amortized posterior approximation*, which allows us to efficiently evaluate approximate posteriors at multiple inputs $x$.

In our case, in the context of a new input $x$, the true posterior distribution over predictors is Multinomial $p(J = j \mid \mathcal{D}, x) = \rho_j(x)$ for $j \in \{1, \ldots, m\}$, where each $\rho_j(x) > 0$ and $\sum_{j=1}^{m} \rho_j(x) = 1$. Thus, we set the variational family to be the set of all multinomial distributions.

$$\mathcal{Q}_x := \{q = (q(J = 1; x), \ldots, q(J = m; x) \in \Delta^{m-1}\}. \tag{18}$$

For a given input $x$, our goal is to minimize the KL divergence

$$\min_{q \in \mathcal{Q}_x} D_{\mathrm{KL}}(q \| p) := \sum_{j=1}^{m} q(J = j; x) \log \frac{q(J = j; x)}{p(J = j \mid \mathcal{D}, x)}. \tag{19}$$

---

**Algorithm 1** IA-BMA: Amortized Posterior Learning (IA-BMA)

---

1: **Inputs:** Training data $\mathcal{D}$; predictors $\{f_j\}_{j=1}^m$; initialization $\theta_0$; learning rate $\eta$; iterations $K$.
2: **Precompute:** For all $i = 1, \ldots, n$ and predictor $j = 1, \ldots, m$, store $\log f_j(y_i \mid x_i)$.
3: **for** $k = 1$ **to** $K$ **do**
4:    **for** $i = 1$ **to** $n$ **do**
5:       **for** $j = 1$ **to** $m$ **do**
6:          **Prior:** Compute $p(J = j | x_{-i}, x) \propto \exp\big(E(J = j; x_{-i}, x_i)\big)$
7:          **Posterior:** Compute $h_{\theta_{k-1}}(x) = (q_{\theta_{k-1}}(J = 1; x_i), \ldots, q_{\theta_{k-1}}(J = m; x_i))$
8:          **ELBO:** Compute

$$\mathcal{L}(x_i; \theta_{k-1}) = \sum_{j=1}^m q_{\theta_{k-1}}(J = j; x_i) \log f_j(y_i \mid x_i) - \sum_{j=1}^m q_{\theta_{k-1}}(J = j; x_i) \log \frac{q_{\theta_{k-1}}(J = j; x_i)}{p(J = j | x_{-i}, x_i)}.$$

9:       **end for**
10:       **Update:** $\overline{\mathcal{L}}(\theta_{k-1}) \leftarrow \frac{1}{n} \sum_i \mathcal{L}(x_i; \theta_{k-1})$
11:    **end for**
12:    **Update:** $\theta_k \leftarrow \theta_{k-1} + \eta \nabla_\theta \overline{\mathcal{L}}(\theta_{k-1})$
13: **end for**
14: **Return:** $\hat{\theta} := \theta_K$

---

Note that since the true posterior and the variational family share the same (categorical) form, the problem is well-specified: the KL depends only on estimating the probabilities $P(J = j; x)$. In particular, the variational posterior can recover the true posterior exactly, up to limitations stemming from access to finite data.

### 3.3 OPTIMIZATION

To minimize the KL divergence in Equation 19, we optimize the evidence lower bound (ELBO) on the log-likelihood (Kingma & Welling, 2014; Rezende & Mohamed, 2015; Blei et al., 2017). We parameterize the variational distribution with a neural network with weights $\theta$, producing $h_\theta(x) = (q_\theta(J = 1; x), \ldots, q_\theta(J = m; x))$, and optimize $\theta$ rather than the output directly. Thus, our objective to fit the amortized posterior is

$$\mathcal{L}(\theta; x) = \mathbb{E}_{q_\theta}[\log p(y \mid x, f_j)] - D_{\mathrm{KL}}(q_\theta \,\|\, p(J | x_{1:n}, x)) \tag{20}$$

$$= \sum_{j=1}^m [q_\theta(J = j; x) \log f_j(y \mid x)] - \sum_{j=1}^m q_\theta(J = j; x) \log \frac{q_\theta(J = j; x)}{p(J = j \mid x_{1:n}, x)}. \tag{21}$$

Note that the expected log-likelihood $\mathbb{E}_{q_\theta}[\log p(y \mid x, f_j)]$ reduces to a weighted sum, so no sampling is required to evaluate our objective. The complete optimization procedure is summarized in Algorithm 1.

Under the Bayesian formulation, the posterior predictive distribution in Eq. 3 yields Bayes-optimal uncertainty. Any deviation of IA-BMA from this ideal arises only from the variational approximation—specifically from the expressiveness of the variational family, and quantified by the KL term in Eq. 19.

**Weight assignment:** After training is complete (see Algorithm 1), with the estimate $\hat{\theta}$, for a new input $x$ we compute $(q_{\hat{\theta}}(J = 1; x), \ldots, q_{\hat{\theta}}(J = m; x))$ and assign $\alpha_j(x) = q_{\hat{\theta}}(J = j; x)$. This yields a predicted value $\hat{p}_\alpha(y \mid x) = \sum_{j=1}^m \alpha_j(x) f_j(y \mid x)$.

## 4 EXPERIMENTS

Our method operates on a fixed pool of pre-trained predictors, rather than learning experts jointly. As a result, the absolute scale or capacity of each predictor is inconsequential, and methods that rely on ensembles or require joint training of experts are not comparable. For approaches such as MoE and DDP, which typically train both experts and gating weights simultaneously, we evaluate variants that are applied on top of the same pre-trained experts.

Specifically, we compare IABMA against (a) non-adaptive baselines: (i) best single predictor, (ii) uniform average over predictors, (iii) accuracy-weighted average, and (iv) classical Bayesian model averaging (BMA); and (b) adaptive methods: (i) Mixture of Experts (MoE) (Jacobs et al., 1991), (ii) Dynamic Local Accuracy (DLA) (Woods et al., 1997), (iii) Synthetic Model Combination (SMC) (Rasmussen & Ghahramani, 2001), (iv) Bayesian Hierarchical Stacking (BHS) (Yao et al., 2022), and (vii) dependent Dirichlet process (DDP) with fixed "atoms" to the pre-trained predictors.

In each experiment we train the candidate predictors, fit the averaging methods on the training set, and evaluate their predictive distributions on the test set.

We conduct extensive evaluation across (i) two synthetic benchmarks, including scale and sensitivity studies, (ii) two large heteroskedastic real-world tasks (personalized medication and credit fraud), and (iii) four UCI benchmarks, to verify that adaptivity does not degrade performance in low-heteroskedastic settings.

Hyperparameters for our method and all baselines were tuned via binary search to maximize average performance (accuracy for classification, RMSE for regression) on a held-out repetition excluded from the analysis. The selected values and further implementation details are provided in Appendix D, with additional data processing and predictor specifications in Section C. Code to reproduce all results is included with the submission and will be released publicly upon acceptance.

## 4.1 SIMULATIONS

### 4.1.1 LINEAR–CIRCULAR HYBRID CLASSIFICATION

We evaluate IA-BMA on a two-dimensional binary task composed of two heterogeneous subpopulations. Half of the samples follow a linear decision rule and are drawn from a Gaussian cluster near $(-1, 0)$; the other half lie on a ring around $(1, 0)$ and follow a circular rule $y = \mathbb{1}r < 1$. We use $n_{\text{train}} = 1000$, $n_{\text{test}} = 500$, and train all methods on the raw coordinates $(x_1, x_2)$.

This construction yields three regions: (i) points linearly separable, (ii) points circularly separable, and (iii) an intermediate overlap where the correct predictor switches. Ideal weighting places mass on linear models in (i), on circular models in (ii), and mixes softly in (iii).

All methods share the same pool of base predictors: polynomial logistic regression (degree 2 and 3), LDA, and two "soft-circle" classifiers based on radial distance to a learned center. Additional details are provided in section C.1

**Results:** Figure 2 shows that IA-BMA achieves highest accuracy and lowest ECE compared to all non-adaptive baselines, as well as all adaptive methods.

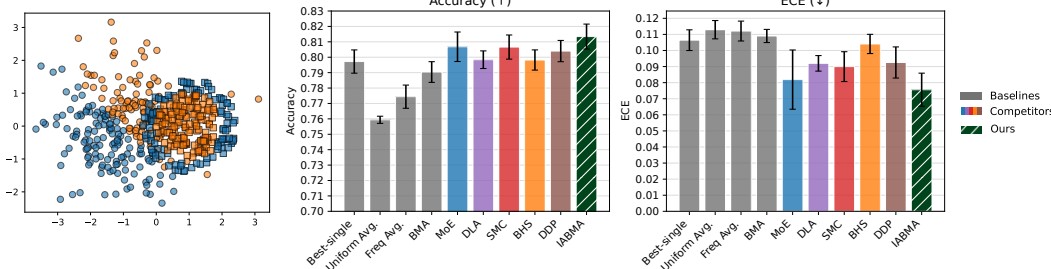

Figure 2: Simulation. Left: data (of one repetition). Results for accuracy (middle) and ECE (right) are reported for 10 repetitions. IA-BMA achieves highest accuracy and lowest ECE.

### 4.1.2 SCALE AND SENSITIVITY ANALYSIS

We evaluate IABMA with respect to: (i) scalability in data dimension, (ii) the number of informative (non-noise) features, (iii) the number of predictors, and (iv) the similarity between predictors.

As detailed in Appendix C.2, we construct a synthetic setting with two data regimes that share only a subset of the informative features. We vary the data dimension $d$, and the number of informative

features $k$. We construct $m$ candidate predictors: (i) two per-regime *specialists*, $m - 3$ *generalists* of varying similarity to each other $\rho$, and an additional model that exceeds all generalists on both regimes but remains inferior to the specialists.

For any input $x$, the optimal ensemble behavior is to select the specialist corresponding to the sign of $x$, and never to select one of the suboptimal generalists or the overall-best predictor.

**Results:** IABMA consistently outperforms all baselines and selects the correct specialist far more often. Table 1 reports specialist selection rates; performance metrics and SDs in Tables 2-6.

Table 1: Correct specialist proportion across scaling experiments.

| Experiment | MoE | DLA | SMC | BHS | DDP | IABMA |
|---|---|---|---|---|---|---|
| Base: $d = 100$, $k = 30$, $m = 10$, $\rho = 0.0$ | 0.000 | 0.008 | 0.051 | 0.037 | 0.227 | **0.948** |
| Dimension increase: $d = 300$ | 0.000 | 0.018 | 0.000 | 0.020 | 0.000 | **0.552** |
| More informative features: $k = 50$ | 0.000 | 0.013 | 0.000 | 0.022 | 0.000 | **0.675** |
| More predictors: $m = 100$ | 0.493 | 0.006 | 0.002 | 0.114 | 0.000 | **0.857** |
| Higher similarity: $\rho = 0.5$ | 0.000 | 0.001 | 0.000 | 0.022 | 0.000 | **0.922** |

## 4.2 CASE STUDIES

### 4.2.1 PERSONALIZED CANCER DRUG-RESPONSE

An important example of heterogeneous data is personalized drug response prediction, where different models may perform better on different subpopulations. We evaluate IA-BMA on this task using the PRISM cancer drug response dataset. The data consists of pairings of molecule-cell line RNA sequence features. For each drug–cell pair we form a continuous response $y$ so that larger values indicate greater sensitivity. We retain drugs with broad site coverage and construct inputs from the top variance genes. All averaging methods operate over the same four base regressors—Ridge, Histogram-based Gradient Boosting Tree, XGBoost, and a Multilayer perceptron (MLP), each with pre-processing tailored to model class. Additional details are provided in C.3.

**Results:** Figure 3 shows that IA-BMA achieves higher $R^2$ and lower RMSE compared to all other methods. Further analysis is presented in Figures 4–7 which display the weights assigned by each averaging method for randomly selected inputs. The results show that IA-BMA consistently favored the best (or nearly best) model, whereas other methods leaned toward other predictors, with MoE in particular overemphasizing MLP and XGB even when suboptimal.

### 4.2.2 CREDIT-CARD FRAUD DETECTION

Another domain characterized by heterogeneous data is fraud detection, where the rarity of fraudulent cases poses an additional challenge. We evaluate IA-BMA on this task using the IEEE-CIS Fraud Detection dataset. The dataset consists from mixed Continuous (such as transaction amount) and high-cardinality categorical features (such as product category), and the target variable $y \in \{0, 1\}$ indicated where a transaction was fraud. All averaging methods operate over the same base classifiers: Logistic Regression with Lasso penalty, Histogram-based Gradient Boosting Tree, XGBoost, and an MLP. Additional details appear in Appendix C.4.

**Results:** Figure 3 shows that IA-BMA achieves higher accuracy and lower expected-calibration error compared to all other methods. Since in fraud prediction calibration matters within each bin, we analyzed per-bin confidence $|p - 0.5|$, and found that IA-BMA achieves the lowest error in all high-confidence bins ($> 0.25$). The corresponding analysis is shown in Figure B.4.

## 4.3 EXPERIMENTS ON UCI BENCHMARK DATASETS

We evaluate IA-BMA on four UCI datasets — two classification (`spambase`, `credit-g`) and two regression (`bike-sharing`, `california-housing`) — which represent *low-heteroskedasticity scenarios*. In such settings, one should not expect consistent dominance by any method, as the benefits of adaptive model averaging emerge when subpopulations differ substan-

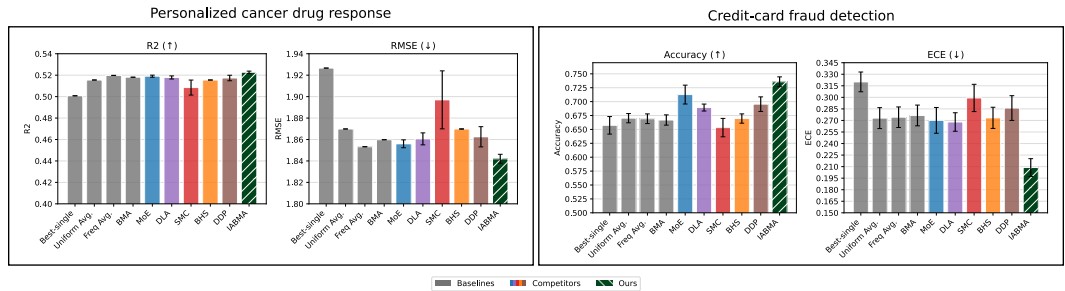

Figure 3: Experimental results for main case studies. Results are reported for 10 repetitions. IA-BMA achieves best results compared to all other averaging method on both case studies.

tially. The value of these experiments lies in confirming that IA-BMA is *non-harmful* in settings where adaptivity is not expected to yield significant gain.

**Results:** IA-BMA yields improvements in RMSE for both regression tasks and in accuracy for both classification tasks, with 9 of 20 pairwise comparisons showing statistically significant gains over five baselines. Experimental details appear in Section C.5. Results are reported in Table 8.

## 5 CONCLUSION

We introduced IA-BMA, a framework that casts model averaging as probabilistic model selection conditioned on the input. Within this formulation, the posterior distribution over models provides the natural, Bayes-optimal choice of input adaptive weights, thereby recovering the true predictive distribution. Our approach is grounded in an input-dependent prior on the selector function and implemented through amortized variational inference of the posterior.

We establish finite-sample bounds showing that the posterior-weights predictor achieves strong likelihood performance compared to any input-specific single-model selector. Empirically, we evaluate IA-BMA across regression and classification tasks, including personalized cancer treatment response, credit-card fraud detection, and standard UCI benchmarks. We show that IA-BMA consistently outperforms both non-adaptive baselines and existing adaptive methods, delivering more accurate and better calibrated predictions.

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

## A PROOFS

### A.1 CHANGE OF MEASURE ARGUMENT

Let $F : X_2 \to X_1$ be a measurable function between two measure spaces $(X_1, \mathcal{A}_1, \eta)$ and $(X_2, \mathcal{A}_2, \nu)$. Let $g : X_1 \to \mathbb{R}$ measurable function. Recall that the change of variables formula is given by

$$\int_{X_1} g \, dF_{\#}\eta = \int_{X_1} (g \circ F) \, d\eta, \tag{22}$$

where $F_{\#}\eta$ denotes the push-forward of $\eta$ through $F$.

Applying this to our setting, a draw from the posterior $g \sim p(g \mid x, \mathcal{D})$ induces a random index $j(x)$ defined by the relation $g(x) = e_{j(x)}$. Formally, the evaluation map

$$s_x : \mathcal{G} \to \{1, \ldots, m\}, \qquad s_x(g) = j(x),$$

pushes the posterior measure $p(g \mid x, \mathcal{D})$ forward onto a distribution over indices. Using this push-forward, we can rewrite equation 8 as

$$\int_{\mathcal{G}} p(y \mid x, g) \, d\mathbb{P}(g \mid x, \mathcal{D}) = \int_{\mathcal{G}} f_{s_x(g)}(y \mid x) \, d\mathbb{P}(g \mid x, \mathcal{D}) \tag{23}$$

$$= \int_{\{1,\ldots,m\}} f_j(y \mid x) \, d\left(E_{x\#}\mathbb{P}\right)(j \mid x, \mathcal{D}) \tag{24}$$

$$= \sum_{j=1}^{m} f_j(y \mid x) \, p(j \mid x, \mathcal{D}). \tag{25}$$

### A.2 PROOF OF THEOREM 2.1

**Theorem.** *Denote $\mathcal{D}_i := \{(x_t, y_t)\}_{t=1}^{i}$, and consider the posterior weights predictor $\hat{p}_{\alpha}^{(i)}$ assigning $\alpha_j^{(i)}(x) := p(J(x) = j \mid \mathcal{D}_i, x)$ to the $j$-th predictor $f_j$. Assume that $\mathbb{E}[|\log f_j(Y \mid X)|] < \infty$ for all $f_j \in \mathcal{F}$. Then, for any measurable selector $j^* : \mathcal{X} \to \{1, \ldots, m\}$ and any $n \geq 1$,*

$$\frac{1}{n} \sum_{i=1}^{n} \mathbb{E}\left[\log \hat{p}_{\alpha}^{(i)}(y_i \mid x_i, \mathcal{D}_{i-1})\right] \geq \mathbb{E}\left[\log f_{j^*(x)}(y \mid x)\right] + \frac{1}{n} \sum_{i=1}^{n} \mathbb{E}\left[\log \alpha_{j^*(x_i)}^{(i)}(x_i)\right], \tag{26}$$

*where the expectations are taken w.r.t the population distribution $(x, y) \sim p(x, y)$.*

*Proof.* Define the posterior-weights predictor

$$\hat{p}_{\alpha}^{(i)}(y \mid x, \mathcal{D}_{i-1}) = \sum_{j=1}^{m} \alpha_j^{(i)}(x) f_j(y \mid x) \tag{27}$$

For a fixed input $x_i$ and a fixed predictor $f_k$ we have that

$$\log \hat{p}_{\alpha}^{(i)}(y_i \mid x_i, \mathcal{D}_{i-1}) = \log\left(\sum_{j=1}^{m} \alpha_j^{(i)}(x_i) f_j(y_i \mid x_i)\right) \tag{28}$$

$$\geq \log\left(\alpha_k^{(i)}(x_i) f_k(y_i \mid x_i)\right) \tag{29}$$

$$= \log f_k(y_i \mid x_i) + \log \alpha_k^{(i)}(x_i). \tag{30}$$

Taking $\mathbb{E}_{(x,y)\sim p(x,y)}[\cdot \mid x_i, \mathcal{D}_{i-1}]$, since $f_{j^*(x)}(y_i \mid x_i)$ is independent of $\mathcal{D}_{i-1}$,

$$\mathbb{E}\left[\log \hat{p}_{\alpha}^{(i)}(y_i \mid x_i, \mathcal{D}_{i-1})\right] \geq \mathbb{E}\left[\log f_k(y_i \mid x_i)\right] + \mathbb{E}\left[\log \alpha_k^{(i)}(x_i) \mid \mathcal{D}_{i-1}\right]. \tag{31}$$

This holds for any $1 \leq k \leq m$, hence for $k = j^*(x_i)$,

$$\mathbb{E}\left[\log \hat{p}_{\alpha}^{(i)}(y_i \mid x_i, \mathcal{D}_{i-1}) \mid \mathcal{D}_{i-1}\right] \geq \mathbb{E}\left[\log f_{j^*(x)}(y_i \mid x_i)\right] + \mathbb{E}\left[\log \alpha_{j^*(x_i)}^{(i)}(x_i) \mid \mathcal{D}_{i-1}\right]. \tag{32}$$

Taking $\mathbb{E}\left[\cdot \mid \mathcal{D}_{i-1}\right]$, by the law of total expectation,

$$\mathbb{E}\left[\log \hat{p}_\alpha^{(i)}(y_i \mid x_i, \mathcal{D}_{i-1})\right] \geq \mathbb{E}\left[\log f_{j^*(x_i)}(y_i \mid x_i)\right] + \mathbb{E}\left[\log \alpha_{j^*(x_i)}^{(i)}(x_i)\right]. \qquad (33)$$

Averaging over $i$, we get

$$\frac{1}{n}\sum_{i=1}^{n}\mathbb{E}\left[\log \hat{p}_\alpha^{(i)}(y_i \mid x_i, \mathcal{D}_{i-1})\right] \geq \mathbb{E}\left[\log f_{j^*(x_i)}(y_i \mid x_i)\right] + \frac{1}{n}\sum_{i=1}^{n}\mathbb{E}\left[\log \alpha_{j^*(x_i)}^{(i)}(x_i)\right].$$

$\square$

## B ADDITIONAL EXPERIMENTAL RESULTS

In what follows we provide a deeper analysis of the performance of adaptive model averaging methods on the two case-studies.

### B.1 SCALE AND SENSITIVITY ANALYSIS

Table 2: Scaling and sensitivity analysis for $d = 100, k = 30, m = 10, \rho = 0.0$

| Method | Accuracy | ECE | Correct Specialist | Global Predictor | Generalists |
|--------|----------|-----|--------------------|------------------|-------------|
| MoE | 0.904 (0.002) | 0.076 (0.003) | 0.000 (0.000) | 1.000 (0.000) | 0.000 (0.000) |
| DLA | 0.825 (0.007) | 0.100 (0.002) | 0.008 (0.001) | 0.966 (0.005) | 0.015 (0.003) |
| SMC | 0.680 (0.132) | 0.302 (0.155) | 0.051 (0.042) | 0.481 (0.424) | 0.414 (0.339) |
| BHS | 0.821 (0.007) | 0.097 (0.002) | 0.037 (0.034) | 0.164 (0.018) | 0.724 (0.036) |
| DDP | 0.910 (0.008) | 0.061 (0.017) | 0.227 (0.280) | 0.773 (0.280) | 0.000 (0.000) |
| IABMA | **0.919 (0.004)** | **0.026 (0.007)** | **0.948 (0.017)** | 0.037 (0.006) | 0.015 (0.019) |

Table 3: Scaling and sensitivity analysis for $\mathbf{d = 300}, k = 30, m = 10, \rho = 0.0$

| Method | Accuracy | ECE | Correct Specialist | Global Predictor | Generalists |
|--------|----------|-----|--------------------|------------------|-------------|
| MoE | 0.858 (0.003) | 0.117 (0.003) | 0.000 (0.000) | 1.000 (0.000) | 0.000 (0.000) |
| DLA | 0.809 (0.005) | 0.068 (0.004) | 0.018 (0.005) | 0.929 (0.004) | 0.037 (0.001) |
| SMC | 0.816 (0.005) | 0.073 (0.004) | 0.000 (0.000) | 1.000 (0.000) | 0.000 (0.000) |
| BHS | 0.804 (0.005) | 0.067 (0.004) | 0.020 (0.002) | 0.150 (0.005) | 0.633 (0.011) |
| DDP | 0.858 (0.003) | 0.117 (0.003) | 0.000 (0.000) | 1.000 (0.000) | 0.000 (0.000) |
| IABMA | **0.882 (0.004)** | **0.035 (0.001)** | **0.552 (0.007)** | 0.163 (0.007) | 0.285 (0.007) |

Table 4: Scaling and sensitivity analysis for $d = 100, \mathbf{k = 50}, m = 10, \rho = 0.0$

| Method | Accuracy | ECE | Correct Specialist | Global Predictor | Generalists |
|--------|----------|-----|--------------------|------------------|-------------|
| MoE | 0.882 (0.008) | 0.092 (0.008) | 0.000 (0.000) | 1.000 (0.000) | 0.000 (0.000) |
| DLA | 0.805 (0.009) | 0.087 (0.005) | 0.013 (0.003) | 0.960 (0.008) | 0.017 (0.003) |
| SMC | 0.813 (0.009) | 0.093 (0.008) | 0.000 (0.000) | 1.000 (0.000) | 0.000 (0.000) |
| BHS | 0.802 (0.008) | 0.084 (0.004) | 0.022 (0.002) | 0.036 (0.004) | 0.735 (0.013) |
| DDP | 0.882 (0.008) | 0.092 (0.008) | 0.000 (0.000) | 1.000 (0.000) | 0.000 (0.000) |
| IABMA | **0.902 (0.007)** | **0.029 (0.006)** | **0.675 (0.010)** | 0.325 (0.010) | 0.000 (0.000) |

### B.2 FORMAL STATISTICAL TESTS

For the other three heteroskedastic experiments, Table 7 provides p-values for one-sided t-test with Benjamini–Hochberg correction for multiple comparisons (accuracy for classification, RMSE for regression). All resulting values are below 0.054.

Table 5: Scaling and sensitivity analysis for $d = 100, k = 50, \mathbf{m} = \mathbf{100}, \rho = 0.0$

| Method | Accuracy | ECE | Correct Specialist | Global Predictor | Generalists |
|---|---|---|---|---|---|
| MoE | **0.926** (**0.005**) | 0.038 (0.002) | 0.493 (0.009) | 0.507 (0.009) | 0.000 (0.000) |
| DLA | 0.758 (0.010) | **0.027** (**0.005**) | 0.006 (0.001) | 0.955 (0.005) | 0.035 (0.005) |
| SMC | 0.725 (0.067) | 0.104 (0.151) | 0.002 (0.004) | 0.755 (0.355) | 0.240 (0.347) |
| BHS | 0.757 (0.010) | **0.027** (**0.005**) | 0.114 (0.005) | 0.019 (0.002) | 0.867 (0.005) |
| DDP | 0.900 (0.002) | 0.081 (0.004) | 0.000 (0.000) | 1.000 (0.000) | 0.000 (0.000) |
| IABMA | 0.916 (0.004) | 0.028 (0.005) | **0.857** (**0.008**) | 0.091 (0.007) | 0.045 (0.002) |

Table 6: Scaling and sensitivity analysis for $d = 100, k = 50, m = 100, \rho = \mathbf{0.5}$

| Method | Accuracy | ECE | Correct Specialist | Global Predictor | Generalists |
|---|---|---|---|---|---|
| MoE | 0.902 (0.006) | 0.073 (0.006) | 0.000 (0.000) | 1.000 (0.000) | 0.000 (0.000) |
| DLA | 0.797 (0.008) | 0.137 (0.002) | 0.001 (0.001) | 0.996 (0.001) | 0.002 (0.000) |
| SMC | 0.837 (0.008) | 0.163 (0.004) | 0.000 (0.000) | 1.000 (0.000) | 0.000 (0.000) |
| BHS | 0.787 (0.008) | 0.126 (0.006) | 0.022 (0.003) | 0.019 (0.004) | 0.952 (0.003) |
| DDP | 0.902 (0.006) | 0.073 (0.006) | 0.000 (0.000) | 1.000 (0.000) | 0.000 (0.000) |
| IABMA | **0.919** (**0.006**) | **0.022** (**0.005**) | **0.922** (**0.001**) | 0.077 (0.001) | 0.000 (0.000) |

p-values for UCI experiments appear in Table 9.

Table 7: Paired t-test p-values vs. IABMA on heterogeneous datasets.

| Experiment | BMA | MoE | DLA | SMC | BHS | DPP |
|---|---|---|---|---|---|---|
| Cancer | < 0.01 | < 0.01 | < 0.01 | < 0.01 | < 0.01 | < 0.01 |
| Fraud | < 0.01 | < 0.01 | < 0.01 | < 0.01 | < 0.01 | < 0.01 |
| Linear-circular | < 0.01 | 0.054 | < 0.01 | 0.037 | < 0.01 | < 0.01 |

## B.3 CANCER TREATMENT RESPONSE

To illustrate how different methods allocate weights, we sampled 16 cases as follows: for each classifier $f_j$, we randomly selected four examples from those where IA-BMA assigned the highest weight to $f_j$. Figures 4–7 display the weights assigned by each averaging method for Ridge, XGB, HGB, and MLP. For each case, we also report the RMSE achieved by the individual classifiers. This analysis shows that in all cases, IA-BMA places the largest weight on the model with either the lowest error or a near-tied second. By contrast, competing methods tend to favor other predictors. In particular, MoE consistently prioritizes MLP or XGB, even in instances where these models are locally suboptimal.

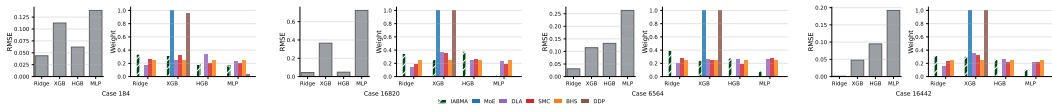

Figure 4: Cases where IA-BMA assigns the highest weight to Ridge.

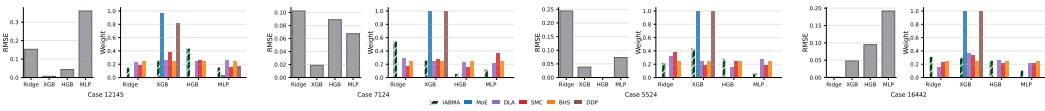

Figure 5: Cases where IA-BMA assigns the highest weight to XGB.

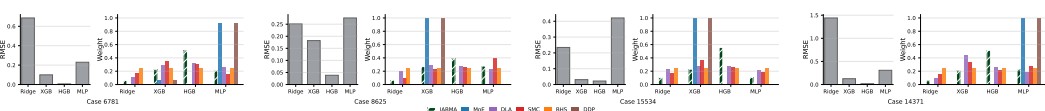

Figure 6: Cases where IA-BMA assigns the highest weight to HGB.

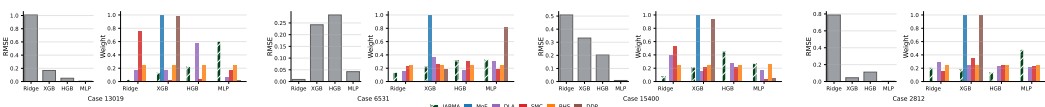

Figure 7: Cases where IA-BMA assigns the highest weight to MLP.

## B.4 CREDIT CARD FRAUD

Credit card fraud prediction is a highly sensitive area, with risks of false alarms and misreporting, calibration is crucial not only overall but also within each bin. To this end, we analyzed the confidence measure $|p - 0.5|$ where p is the estimated probability, which captures certainty for both positive and negative events, and compared the bin-wise errors across averaging methods. Figure B.4 shows that in all bins, IA-BMA attains the lowest error, decreasing with confidence, showing that most wrong predictions occur in low confidence instances.

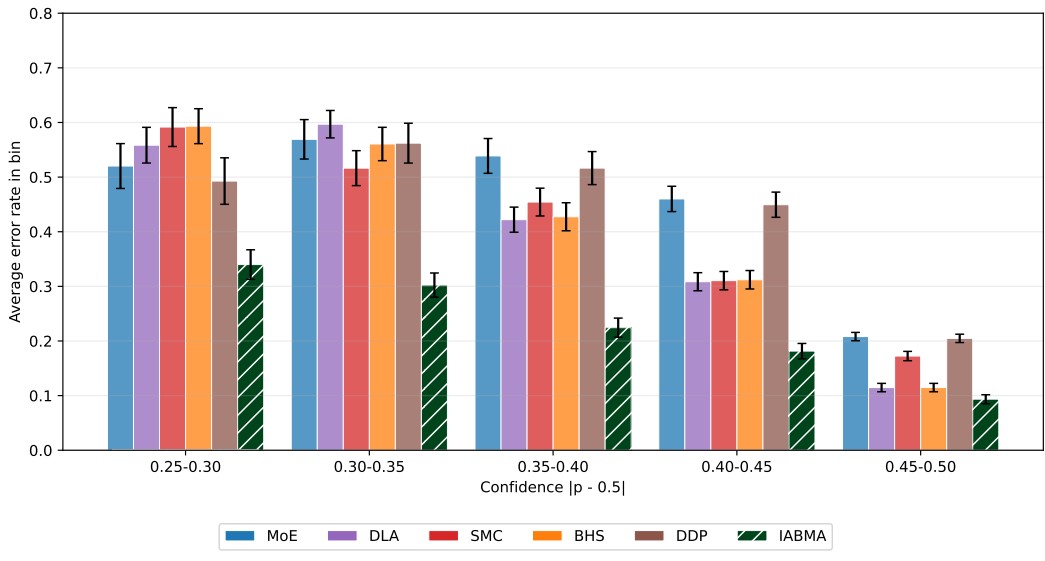

Figure 8: Calibration across confidence bins in credit-card fraud prediction

## B.5 UCI BENCHMARK DATASETS

Results are reported in Table 8, and p-values for one-sided t-test results with Benjamini–Hochberg correction for multiple comparisons (accuracy for classification, RMSE for regression) in Table 9.

## B.6 ANALYSIS OF THE EFFECT OF THE ADAPTIVE PRIOR

IA-BMA places an input–dependent prior over selectors $g$. Therefore, variability in our model arises not only from variability of $g(x)$ as in other adaptive methods, but additionally from the adaptivity

Table 8: UCI benchmarks: mean (sd) across runs.

| Dataset | Metric | Best single | Uniform Avg. | Freq Avg. | BMA | MoE | DLA | SMC | BHS | DDP | IA-BMA |
|---|---|---|---|---|---|---|---|---|---|---|---|
| Bike-sharing | R2 ($\uparrow$) | 0.706 (0.022) | 0.752 (0.010) | 0.773 (0.012) | 0.774 (0.014) | 0.706 (0.022) | 0.781 (0.013) | 0.756 (0.010) | 0.752 (0.010) | 0.753 (0.013) | **0.794** (0.010) |
| Bike-sharing | RMSE ($\downarrow$) | 0.582 (0.033) | 0.491 (0.021) | 0.448 (0.020) | 0.447 (0.021) | 0.581 (0.033) | 0.446 (0.020) | 0.483 (0.020) | 0.491 (0.021) | 0.479 (0.022) | **0.433** (0.018) |
| Cal.-housing | R2 ($\uparrow$) | 0.772 (0.022) | 0.840 (0.018) | 0.840 (0.017) | 0.812 (0.017) | 0.778 (0.024) | 0.840 (0.018) | 0.817 (0.066) | 0.840 (0.018) | 0.805 (0.031) | **0.844** (0.014) |
| Cal.-housing | RMSE ($\downarrow$) | 0.036 (0.004) | 0.025 (0.003) | 0.025 (0.003) | 0.029 (0.003) | 0.035 (0.004) | 0.025 (0.003) | 0.029 (0.010) | 0.025 (0.003) | 0.031 (0.005) | **0.024** (0.003) |
| Credit-g | Accuracy ($\uparrow$) | 0.634 (0.036) | 0.676 (0.029) | 0.662 (0.038) | 0.648 (0.036) | 0.624 (0.036) | 0.668 (0.039) | 0.626 (0.046) | 0.682 (0.039) | 0.655 (0.040) | **0.684** (0.047) |
| Credit-g | ECE ($\downarrow$) | 0.260 (0.034) | 0.172 (0.022) | **0.169** (0.020) | 0.174 (0.023) | 0.296 (0.035) | 0.173 (0.025) | 0.222 (0.038) | 0.176 (0.25) | 0.257 (0.031) | 0.175 (0.020) |
| Spambase | Accuracy ($\uparrow$) | 0.699 (0.110) | 0.702 (0.094) | 0.738 (0.044) | 0.760 (0.024) | 0.760 (0.035) | 0.729 (0.052) | 0.757 (0.032) | 0.646 (0.132) | 0.754 (0.064) | **0.764** (0.032) |
| Spambase | ECE ($\downarrow$) | 0.114 (0.022) | 0.163 (0.049) | 0.148 (0.042) | 0.169 (0.018) | **0.095** (0.034) | 0.171 (0.051) | 0.222 (0.023) | 0.180 (0.061) | 0.149 (0.042) | 0.146 (0.025) |

Table 9: Paired t-test p-values vs. IA-BMA on UCI benchmarks.

| **Experiment** | BMA | MoE | DLA | SMC | BHS |
|---|---|---|---|---|---|
| Credit-g | 0.104 | < **0.01** | 0.338 | < **0.01** | 0.758 |
| Spambase | 0.349 | 0.380 | **0.019** | 0.349 | **0.016** |
| Bike-sharing | 0.316 | < **0.01** | 1.000 | < **0.01** | < **0.01** |
| California-housing | < **0.01** | < **0.01** | 0.153 | 0.153 | 0.153 |

of the prior itself. To isolate the contribution of this adaptive prior, we evaluate our method under a uniform prior and compare the resulting behavior.

The results below (Tables 10 and 11) show that without the adaptive prior, our method still outperforms other methods in most cases, but with a smaller margin.

Table 10: Impact of the input–adaptive prior on RMSE across regression tasks

| Experiment | Best-single | Uniform Avg | Freq Avg | BMA | MoE | DLA | SMC | BHS | DDP | IA-BMA | IA-BMA Uniform |
|---|---|---|---|---|---|---|---|---|---|---|---|
| PRISM | 1.927 | 1.870 | 1.853 | 1.860 | 1.856 | 1.861 | 1.897 | 1.870 | 1.863 | 1.842 | 1.853 |
| Bike-Sharing | 0.582 | 0.491 | 0.448 | 0.447 | 0.581 | 0.433 | 0.483 | 0.491 | 0.479 | 0.446 | 1.311 |
| California-Housing | 0.036 | 0.025 | 0.025 | 0.029 | 0.035 | 0.025 | 0.029 | 0.025 | 0.031 | 0.024 | 0.019 |

Table 11: Impact of the input–adaptive prior on accuracy across classification tasks

| Experiment | Best-single | Uniform Avg | Freq Avg | BMA | MoE | DLA | SMC | BHS | DDP | IA-BMA | IA-BMA Ablation |
|---|---|---|---|---|---|---|---|---|---|---|---|
| Fraud small | 0.657 | 0.670 | 0.669 | 0.667 | 0.713 | 0.689 | 0.653 | 0.670 | 0.695 | 0.736 | 0.713 |
| Synthetic Binary | 0.797 | 0.759 | 0.774 | 0.790 | 0.807 | 0.798 | 0.807 | 0.798 | 0.804 | 0.813 | 0.813 |
| Credit-G | 0.634 | 0.684 | 0.662 | 0.648 | 0.624 | 0.668 | 0.626 | 0.682 | 0.655 | 0.676 | 0.575 |
| Spambase | 0.699 | 0.702 | 0.738 | 0.760 | 0.760 | 0.729 | 0.757 | 0.646 | 0.754 | 0.764 | 0.739 |

# C  EXPERIMENTAL DETAILS

## C.1  LINEAR–CIRCULAR HYBRID CLASSIFICATION

### C.1.1  DATA AND PROCESSING

We generated a two-dimensional binary dataset with two subpopulations governed by different decision rules. For the *linear* subpopulation, we drew $n_{\text{lin}} = n_{\text{train}}/2$ training points from a Gaussian

cloud centered at $(-t, 0)$ (with $t = 1$),

$$X^{(\text{lin,train})} \sim \mathcal{N}\big((-t, 0),\, 0.1\, I_2\big),$$

and assigned labels by a linear rule $y = \mathbb{1}\{x_1 + x_2 > -t\}$. For the *circular* subpopulation, we drew $n_{\text{circ}} = n_{\text{train}} - n_{\text{lin}}$ points on a ring around $(t, 0)$ by sampling $\theta \sim \text{Unif}(0, 2\pi)$ and $r = \sqrt{U}$ with $U \sim \text{Unif}(0, 2)$, and set

$$X^{(\text{circ})} = (t, 0) + \big(r \cos \theta,\, r \sin \theta\big), \qquad y = \mathbb{1}\{r < 1\}.$$

We used $n_{\text{train}} = 1,000$ and $n_{\text{test}} = 500$; the train/test splits were generated independently.

Only the two coordinates $(x_1, x_2)$ were provided as features. A region indicator $z \in \{0 \text{ (linear)},\, 1 \text{ (circular)}\}$ was recorded for analysis but was not used during training.

This dataset naturally forms three regimes: (i) linearly separable points, (ii) circularly separable points, (iii) an ambiguous overlap region where neither boundary dominates.

### C.1.2  CANDIDATE PREDICTORS

All averaging methods were evaluated on the same 3 base classifiers:

1. *Polynomial logistic regression (degrees 2 and 3).* We fit logistic regression with polynomial features of degree $d \in \{2, 3\}$ (no bias term in the expansion).

2. *Linear Discriminant Analysis (LDA).* A linear generative classifier fit on the raw coordinates, providing a single linear boundary.

3. *Soft-circle classifiers (two instances).* Each instance modeled the positive-class probability as a logistic function of radial distance to a fixed center,

$$p_{\text{circle}}(y{=}1 \mid x) = \sigma\big(\gamma\,(R - \|x - c\|)\big), \quad c = (0.8\, t, 0),\ R = 1.0,\ \gamma = 5.0,$$

   yielding smooth circular decision regions around $(t, 0)$.

   We include two instances of each predictor to allow the averaging procedure to allocate weight among near-identical experts.

## C.2  SCALE AND SENSITIVITY ANALYSIS

We evaluate IABMA with respect to four factors: (i) scalability in data dimension, (ii) the number of informative (non-noise) features, (iii) the number of predictors, and (iv) the similarity between predictors.

### C.2.1  DATA

We generate inputs $x \in \mathbb{R}^d$ by sampling each coordinate independently from $0, 1$ with probability $1/2$. With probability $1/2$ we flip the sign of the entire vector, creating two regimes: positive and negative. Labels $y$ are assigned by majority vote over $k$ designated coordinates. The identity of these coordinates differs across regimes, and we control the fraction of shared coordinates $\rho$ to control heteroskedasticity. Independent Gaussian noise $\mathcal{N}(0, 0.1)$ is then added to each input, identically for training and test sets.

### C.2.2  CANDIDATE PREDICTORS

We form a pool of $m$ logistic predictors. Two *specialists* are trained exclusively on one regime each, performing well on that regime, and poorly on the other. The remaining $m - 3$ *generalists* are trained on mixtures of the two regimes, yielding weaker per-regime accuracy; varying the mixture proportion $p$ controls their similarity.

Finally, we include a two-layer MLP (32 and 16 units, ReLU activations) trained on combined balanced data. It is designed to exceed all generalists on both regimes, but remains inferior to the specialists.

For any input $x$, the optimal ensemble behavior is to select the specialist corresponding to the sign of $x$, and never to select one of the suboptimal generalists or the overall-best predictor.

We set our baseline experiment parameters as $d = 100, k = 30, \rho = 0.0, m = 11$. We then vary each.

## C.3 PRISM CANCER EXPERIMENT

### C.3.1 DATA AND PROCESSING

We used the publicly available PRISM cancer drug response dataset. The primary data[3] was combined with an RNA-seq expression matrix[4], cell-line metadata[5], and tissue labels[6]. All files are available from https://depmap.org/portal/data_page/.

The PRISM file reports drug–cell line responses with identifiers of the form ACH-#. We normalized all identifiers to the canonical zero-padded format (ACH-XXXXXX). Non-Continuous entries and all observations lacking a primary cancer site were excluded. Responses correspond to log-fold changes (LFC), clipped to the range $[-6, 6]$, and the prediction target was defined as $y = -v$, where $v$ is the clipped LFC.

We focused on the 40 drugs with the greatest site-level heterogeneity. Specifically, we computed the between-site variance of $y$ and retained compounds observed in at least 3 distinct sites, with at least 5 samples per site and at least 40 samples overall. A minimum per-site coverage threshold of 20 samples was enforced. To avoid domination by a few large tissues, we capped each site at $1.1 \times s$, where $s$ is its sample size. This yielded approximately $18{,}460$ drug–cell line pairs (slight variation across random splits), of which $80\%$ were used for training and $20\%$ for testing.

Gene expression features were restricted to the 100 highest-variance genes. Each gene was standardized to mean 0 and variance 1 based on training statistics. The final feature matrix consisted of standardized gene expression values and a categorical compound indicator.

The full processing code was submitted with this paper and will be released publicly upon acceptance.

### C.3.2 CANDIDATE PREDICTORS

All averaging methods were evaluated on averaging the same four regression models with reprocessing pipelines tailored per model:

1. *Ridge regression ($\ell_2$ regularized linear model).* Gene features were imputed (median), standardized to zero mean and unit variance, and combined with a dense one-hot encoding of the compound identity.

2. *Histogram-based Gradient Boosting regressor (HGB).* Tree-based model trained on raw gene values (median imputation only) together with a sparse one-hot encoding of the compound identity.

3. *XGBoost regressor (XGB).* Gradient-boosted decision trees with squared-error objective, trained using the same pre-processing as HGB. We used 400 estimators, learning rate 0.05, maximum depth 8, subsample ratio 0.9, and column subsample ratio 0.8, with $\ell_1$ and $\ell_2$ regularization.

4. *Multi-layer perceptron (MLP).* A feed-forward neural network with hidden layers of size $(128, 64)$, ReLU activations, learning rate $10^{-3}$, batch size 64, and early stopping based on a 10% validation split. Inputs were preprocessed as for Ridge (dense, imputed, standardized gene features and dense one-hot drug encoding).

---

[3]Repurposing_Public_23Q2_Extended_Primary_Data_Matrix.csv

[4]OmicsExpressionProteinCodingGenesTPMLogp1.csv

[5]Cell_lines_annotations_20181226.txt

[6]Model.csv

## C.4 IEEE-CIS FRAUD EXPERIMENT

### C.4.1 DATA AND PROCESSING

We used the IEEE-CIS credit-card fraud dataset, available at https://www.kaggle.com/c/ieee-fraud-detection/data.

We removed rows with missing target (isFraud) and features with more than $50\%$ missing values. To limit explosion in feature dimension, infrequent categories were grouped into a shared rare category.

In each repetition $80\%$ of the data was used for training and $20\%$ for testing. The training data was then reduced to obtained class balance, while in test data class imbalance was maintained. To reduced covariate shift in the train-test split we stratified jointly on (ProductCD, card4) crossed with per-row missingness bins and TransactionAmt quantile bins, with a fallback "RARE" bucket for very small strata. This procedure yielded a stable empirical mix of products, card networks, and spending levels. Specifically, to control the empirical mix of products, card networks, and spending levels we stratified jointly on (ProductCD, card4) crossed with per-row missingness bins and TransactionAmt quantile bins.

Continuous features were median-imputed and where appropriate, standardized to zero mean and unit variance. Categorical features were imputed to the most frequent level and one-hot encoded, with infrequent categories pooled into a rare-level. Class imbalance was addressed within each classifier as noted below.

### C.4.2 CANDIDATE PREDICTORS

All averaging methods were evaluated over the same following base classifiers.

1. *Logistic Regression ($\ell_1$-penalized).* We fit a penalized logistic model to the processed feature set, using an $\ell_1$ penalty with strength to encourage sparsity and robustness to correlated predictors. We used a saga solver, $\ell_1$ penalty with regularization strength of $0.05$, maximal number of iterations as $4000$, and tolerance of $10^{-3}$.

2. *XGBoost (XGB).* We trained a gradient-boosted ensemble of shallow decision trees using histogram-based splits and early stopping. Depth, learning rate, and number of estimators were selected via a held-out validation set. Hyper parameters were set as maximal bin of $256$, $300$ estimators, maximal depth of $5$, learning rate $0.1$, row subsampling of $0.3$, feature subsampling of $0.7$, and $\ell_2$ penalty with strength $1.0$.

3. *Histogram-based Gradient Boosting (HGB).* We train boosted trees with a histogram grow policy, subsampling of observations and features, and $\ell_2$ regularization. Class imbalance was addressed via the standard positive-class weight $\frac{n_{\text{neg}}}{n_{\text{pos}}}$, estimated from the training examples. Hyperparameters (learning rate, depth, estimators, subsampling ratios) were fixed based on validation performance and kept constant across comparisons. Hyperparameters were set to maximal depth of $4$, learning rate $0.07$, and $\ell_2$ regularization with strength $0.5$, and at most $350$ iterations.

4. *Multi-layer perceptron (MLP).* We used a feed-forward network with two hidden layers of sizes $384$, $192$ and ReLU activations, trained with weight decay and early stopping on a validation split. Weight decay was set to $\alpha = 3 \cdot 10^{-3}$, batch size $512$, adaptive learning rate with initial value of $10^{-3}$, early stopping with validation fraction $0.12$ and no change for $12$ iterations, maximal number of iterations as $300$, and tolerance $10^{-4}$.

## C.5 UCI EXPERIMENTS

### C.5.1 DATA AND PROCESSING

We evaluated IA-BMA on standard UCI tasks retrieved from OpenML. We chose datasets with relatively large number of observations and features. For *classification*, we used spambase (target: class) and credit-g (target: class). For *regression*, we used bike-sharing (target: cnt) and california-housing (target: MedHouseVal).

We replaces common "unknown" tokens (e.g., `?`, `NA`, `NaN`, `unknown`) with missing values, stripping whitespace on string columns in each dataset, and dropped features whose missing rate exceeded $40\%$.

We used an $80\%/20\%$ train-test split in each repetition. For classification, we performed stratified sampling on the label to preserve class proportions in the test set, and then balanced only the training split by downsampling the majority class to the minority size. For regression, we created an approximately balanced split by binning the continuous target into 12 quantile bins and stratifying on those bins. All pre-processing statistics (imputation, scaling, and one- hot vocabularies) were computed on the training partition and applied unchanged to the test data.

To encourage diversity among base models, we formed several heterogeneous, partially overlapping *feature bundles* and trained each model on a bundle tailored to its strengths. Bundles were constructed from the training data as follows:

- **B1:** up to 3 Continuous features with highest absolute Pearson correlation with the target (continuous median-imputed for this computation).

- **B2:** up to 3 highest-variance Continuous features.

- **B3:** up to 3 categorical features with highest cardinality.

- **B4:** up to 5 remaining low-cardinality categorical variables.

- **B5:** all categorical features.

- **B6:** all Continuous features.

- **B7:** the union of **B1** and **B3**.

Continuous features in non-tree models were median-imputed and standardized. Categorical features were imputed to the most frequent level and one-hot encoded with a minimum frequency threshold of 10 to pool rare levels; unknown categories at test time were ignored.

### C.5.2 CANDIDATE PREDICTORS

Across all datasets we trained a common set of base learners. For classification: Multinomial Naive Bayes, $k$–NN ($k = 3$), Random Forest, Extra Trees, and a linear SVM. For regression: Ridge ($\alpha$=0.05), Lasso ($\alpha$=0.05), $k$–NN ($k$=3, distance-weighted), Random Forest, and Extra-Trees. To encourage diversity, each model was trained on a subset of features ("feature bundles").

## D IMPLEMENTATION DETAILS

In all our experiments the posterior network for IA-BMA and the gating network for MoE were implemented as feed-forward neural networks with hidden layers of size (64, 32, 16) and ReLU activations. We used Adam optimizer for MoE and IA-BMA across all experiments.

Hyperparameters for our method and all baselines were tuned via binary search to maximize average performance (accuracy for classification, RMSE for regression) on a held-out repetition excluded from the analysis. The selected values and running times by experiment and method are reported below.

### D.1 HYPERPARAMETERS OF ENSEMBLE METHODS

Table 12: Hyperparameters of Mixture-of-Experts

| Hyperparameter | Synthetic | PRISM (Cancer) | Fraud (IEEE-CIS) | UCI |
|---|---|---|---|---|
| Learning rate | $10^{-3}$ | $10^{-3}$ | $10^{-3}$ | $10^{-3}$ |
| Batch size | 64 | 128 | 64 | 64 |
| Epochs | 10 | 20 | 10 | 10 |

Table 13: Hyperparameters of Dynamic Local Accuracy (DLA).

| Hyperparameter | Synthetic | PRISM (Cancer) | Fraud (IEEE-CIS) | UCI |
|---|---|---|---|---|
| Neighborhood size $k$ | 50 | 50 | 50 | 50 |
| Temperature $T$ | 0.8 | 1.0 | 1.0 | 1.0 |
| Smoothing $\alpha$ | 1.0 | 1.0 | 1.0 | 1.0 |

Table 14: Synthetic Mixture of Experts (SMC).

| Hyperparameter | Synthetic | PRISM (Cancer) | Fraud (IEEE-CIS) | UCI |
|---|---|---|---|---|
| Confident-cover threshold | 0.6 | 0.6 | 0.6 | 0.6 |
| Cover quantile (reg.) | – | 0.30 | – | 0.30 |
| Min coverage per model | 20 | 20 | 20 | 20 |
| Cov. reg. (reg. mix) | 0.9 (Gaussian scores) | 0.9 | 0.9 | 0.7 |

Table 15: Bayesian Hierarchical Stacking (BHS).

| Hyperparameter | Synthetic | PRISM (Cancer) | Fraud (IEEE-CIS) | UCI |
|---|---|---|---|---|
| Temperature $T$ | 1.0 | 1.0 | 1.0 | 1.0 |
| Prior weight | 1.0 | 1.0 | 1.0 | 1.0 |
| Slab scale $s_0$ | 5.0 | 5.0 | 5.0 | 5.0 |
| Learning rate | $5 \times 10^{-3}$ | $5 \times 10^{-3}$ | $5 \times 10^{-3}$ | $10^{-3}$ |
| Batch size | 64 | 128 | 64 | 64 |
| Epochs | 10 | 20 | 10 | 10 |

Table 16: Input Adaptive Bayesian Model Averaging (IA-BMA)

| Hyperparameter | Synthetic | PRISM (Cancer) | Fraud (IEEE-CIS) |
|---|---|---|---|
| Learning rate | $10^{-3}$ | $10^{-3}$ | $10^{-3}$ |
| Batch size | 64 | 128 | 64 |
| Epochs | 10 | 30 | 10 |
| KL weight $\lambda_{KL}$ | 0.05 | 0.2 | 0.2 |

Table 17: IA-BMA (PosteriorNet) hyperparameters per UCI dataset.

| Hyperparameter | Spambase (clf) | Credit-g (clf) | Bike-sharing (reg) | Cal housing (reg) |
|---|---|---|---|---|
| Learning rate | $5 \times 10^{-3}$ | $5 \times 10^{-3}$ | $1 \times 10^{-3}$ | $1 \times 10^{-3}$ |
| Batch size | 64 | 64 | 64 | 64 |
| Epochs | 10 | 10 | 10 | 10 |
| KL weight $\lambda_{KL}$ | 0.1 | 0.1 | 0.8 | 3.0 |

# E    RUNTIMES

Overall run-times per method are reported in Table 18. While computational cost scales with number of predictors and data samples, across all experiments run-times of IA-BMA remmain consistent with those of MoE and DDP with the same network architectures.

Table 18: Method runtimes (seconds): mean (sd) across 10 repetitions.

| Experiment | MoE | DLA | SMC | BHS | DPP | IA-BMA |
|---|---|---|---|---|---|---|
| Cancer | 147.359 (5.282) | 22.269 (0.454) | 0.072 (0.114) | 28.572 (1.167) | 271.320 (5.129) | 252.985 (5.571) |
| Fraud | 439.502 (129.487) | 8.246 (1.719) | 688.473 (155.629) | 16.622 (3.139) | 502.854 (12.831) | 461.312 (121.168) |
| Simulation | 5.664 (0.104) | 0.218 (0.008) | 0.079 (0.004) | 1.040 (0.079) | 6.381 (0.075) | 5.889 (0.038) |
| Bike-Sharing | 25.080 (3.780) | 0.868 (0.179) | 0.007 (0.001) | 21.364 (1.063) | 31.889 (3.971) | 29.663 (4.094) |
| Cal. housing | 8.510 (0.987) | 0.350 (0.041) | 0.006 (0.001) | 7.281 (0.324) | 10.785 (1.122) | 9.815 (1.029) |
| Credit-g | 3.178 (0.049) | 0.439 (0.017) | 0.174 (0.007) | 1.184 (0.148) | 3.345 (0.071) | 3.345 (0.048) |
| Spambase | 16.420 (0.287) | 0.642 (0.025) | 0.822 (0.159) | 1.781 (0.147) | 20.831 (5.367) | 18.651 (5.122) |
| Scale ($m=10, d=100$) | 40.977 (3.83) | 1.794 (0.16) | 28.174 (9.61) | 3.009 (0.26) | 48.898 (3.72) | 40.333 (4.12) |
| Scale ($m=10, d=300$) | 40.517 (3.84) | 4.158 (0.17) | 85.922 (8.65) | 2.501 (0.21) | 50.541 (3.68) | 40.650 (3.21) |
| Scale ($m=100, d=100$) | 41.638 (3.36) | 2.169 (0.75) | 254.196 (18.17) | 2.612 (0.23) | 51.914 (3.83) | 45.433 (3.55) |

