# OpenReview forum: "Input-Adaptive Bayesian Model Averaging"
_ICLR.cc/2026/Conference — Submitted to ICLR 2026_

### Official Review · Reviewer_E3vv · 2025-10-31

**Soundness:** 3
**Presentation:** 3
**Contribution:** 3
**Rating:** 4
**Confidence:** 4

**Summary:**

This paper addresses the challenge/difficulty of combining multiple predictive models, which is especially difficult in heterogeneous settings where different models may be optimal for different inputs. Standard model averaging uses a single set of global weights, which performs poorly in these scenarios. To solve this, the authors propose Input-Adaptive Bayesian Model Averaging (i.e. the IABMA method), a Bayesian method that calculates specific weights for each model conditional on the input data $x$.

The IABMA method models the choice of the best model as a random selector function g that depends on the input x. It uses an input-adaptive prior and then calculates a posterior distribution over which the model is most plausible given the training data ($D$) and the specific input $x$. This posterior distribution, estimated using amortized variational inference, directly gives the optimal, input-specific weights for combining the models. The authors derive formal guarantees for the performance of IABMA and then evaluate it on regression and classification tasks, including popular UCI datasets such as personalized cancer treatment and fraud detection. The results demonstrate that IABMA consistently delivers competitive performance to existing non-adaptive  and adaptive methods.

**Strengths:**

One of the main strengths of the paper is that it is clearly written and structured. The authors lay out their argument in a straightforward manner, making the text and the core concepts of the proposed IABMA method relatively easy to follow.

The paper addresses an interesting and relevant problem in model averaging, particularly for heterogeneous data where input-specific models are needed. The proposal itself is presented in a "clean" and simple probabilistic formulation. The paragraphs are well written, the theory is well motivated and the flow of the paper is clear. The intuitive example in section 3.1 as well as the actual taking care of the problem (e.g. how to fit the variational distributions or how to optimize the KL divergence) are well formulated and explained.

Additionally, the authors provide motivation for their framework by connecting it to practical examples like personalized medicine and fraud detection. They also include some theoretical development to support their method, which helps to frame the potential benefits of the approach.

**Weaknesses:**

While the core idea of the paper is clean and simple, this simplicity in my opinion necessitates a much stronger and elaborate set of experiments to fully justify its contribution. The initial motivation and theory are strong, but the paper ultimately falls short due to a weak and insufficiently comprehensive evaluation. This gap between the promising setup and the empirical evidence was a bit disappointing.

In my opinion, the existing experiments are not presented in a convincing manner. For instance, Figures 2 and 3 take a significant amount of space to present results that could be summarized more efficiently in a small table, especially since the paper has some space to spare. More importantly, it is not clear from the results that the authors have obtained if the performance differences between the proposed method and the baselines is statistically significant (it is not readable from the plots). The comparison to other adaptive competitors, such as Mixture of Experts for example, feels underdeveloped. In many cases, the results do not show a lcear advantage for IABMA over others, and the authors missed an opportunity to properly articulate why one would choose their method over existing, well-established alternatives.

Furthermore, the scope of the analysis is quite limited. The authors only consider a small set of candidate models (e.g., four regressors), which leaves several important questions unanswered. A more thorough evaluation could have investigated:
1.  How the method's performance scales with a larger number of candidate models.
2.  How the method performs when several predictors yield similar (or maybe redundant) performance
3.  The magnitude of the performance gain in a setting specifically designed to be highly heterogeneous, where input-adaptive averaging would be expected to clearly outperform input-agnostic methods.

Finally, the paper could be strengthened by doing some rewriting of its analyses and its context. The qualitative analysis in Appendix B.1, for example, is quite interesting and a portion of it should have been included in the main text to provide better insight and overview of the performance. Given the community's shift toward larger datasets, the paper would also benefit from at least one or two analyses on a larger-scale benchmark to demonstrate its relevance and scalability. Also, there is no discussion of computational cost; it is unclear if fitting the amortized variational posterior is more or less costly than maximizing the likelihood in methods like MoE, or how the methods compare in terms of speed.

**Questions:**

Please see weaknesses section.

---

> ### Author Response · Authors · 2025-11-21
> **Response -- part 1**
>
> Dear reviewer,
>
> Thank you for your detailed review.
>
> Precisely due to a shift towards large datasets (where retraining multiple predictors jointly is expensive), our method assumes a fixed set of pre-trained predictors. Most prior adaptive approaches, including MoE, were designed to jointly train both predictors and the gating mechanism (and therefore differ mainly in the assumed predictor class). Thus, the only meaningful **comparison to MoE** is through an adapted variant that operates on top of pre-trained predictors, which is the one we evaluate.
>
> In all previous adaptive methods, adaptivity is achieved through maximizing the likelihood with respect to weights that vary with $x$. This effectively partitions $\mathcal{X}$ into regions dominated by the expert that assigns highest score to the correct label, **regardless of its own possible over-confidence**. In contrast, **an advantage of IABMA** is that the weights are set posterior probabilities of a predictor generating $(x,y)$, and thus correspond to **perfect calibration** (Eq 3).
>
> Our experiments were designed to: (i) gain intuition using clear synthetic data (ii) apply IABMA to **high-heterogeneity real-world tasks** (personalized cancer medication, fraud detection), and (iii) examine IABMA on **low-heterogeneity UCI benchmarks**. Our results show that in practice, the estimated weights in IABMA assign higher weight to predictors that make accurate predictions for the specific input $x$, even if those predictors perform relatively poorly on average across all inputs. To emphasize this, we now added **formal statistical tests** showing that the gains of IABMA in these experiments are statistically significant (please see below).
>
> Regarding **scalability**, please note that our experiments included high-scale data: as described in Appendix C.2, after processing the cancer treatment data consisted of approximately 18,460 observations, and we focused on the 100 most variable genes (features). Similarly, the fraud data consists of 125,000 observations and 218 features. Nevertheless, following your suggestion, we constructed **an additional set of experiments** designed to evaluate scalability to (i) the number of **total features**, (ii) number of **informative features**, (iii) level of **heteroscedasticity**, (iv) **number of predictors**, (v) **redundancy** among predictors. The results (please see below) show that **across all these IABMA performs significantly better than all competing methods**.

---

> ### Author Response · Authors · 2025-11-21
> **Statistical tests for existing experiments**
>
> he following p-values of one-sided t-test results with Benjamini–Hochberg correction for multiple comparisons (accuracy for classification, RMSE for regression). As you can see, **all these comparisons show significant gain**.
>
>  | Experiment       | BMA       | MoE       | DLA       | SMC       | BHS       |
> |:-----------------|:----------|:----------|:----------|:----------|:----------|
> | cancer           | <0.01 | <0.01 | <0.01 | <0.01 | <0.01 |
> | fraud            | <0.01 | <0.01 | <0.01 | <0.01 | <0.01 |
> | synthetic_binary | <0.01 | 0.054 | <0.01  | 0.037 | <0.01 |
>
> The UCI datasets serve as low-heteroskedastic benchmarks, where we would not expect any method to perform consistently better than others. Nevertheless, in these settings, our method improves RMSE in both regression tasks and accuracy in both classification tasks (Credit-G, Spambase). Roughly half of these improvements are statistically significant (9 of 20 pairwise comparisons) (over 5 competitors across 4 UCI datasets).
>
> | Experiment             | BMA       | MoE       | DLA       | SMC       | BHS       |
> |:-----------------------|:----------|:----------|:----------|:----------|:----------|
> | uci-credit-g     | 0.104     | **<0.01** | 0.338     | **<0.01** | 0.758     |
> | uci-spambase     | 0.349     | 0.380     | **0.019** | 0.349     | **0.016** |
> | uci-bike-sharing       | 0.316     | **<0.01** | 1.000     | **<0.01** | **<0.01** |
> | uci-california-housing | **<0.01** | **<0.01** | 0.153     | 0.153     | 0.153     |

---

> ### Author Response · Authors · 2025-11-21
> **Scale Analysis**
>
> We generate inputs $x \in \mathbb{R}^{d}$ by sampling each coordinate independently from $\{0,1\}$ with probability $1/2$. Each sample is then assigned to one of two regimes: with probability $1/2$ we flip the sign of the entire vector, producing a \emph{positive} regime ($x = x_{\mathrm{base}}$) and a \emph{negative} regime ($x = -x_{\mathrm{base}}$).
> The label $y$ is set to the majority vote of $k$ coordinates. The identity of these coordinates is determined by the regime, where we vary the proportion of overlapping coordinates $\rho$ between the regimes to produce different heteroskedasticity levels. Finally, each input is perturbed by independent Gaussian noise, applied identically to both training and test data.
> We construct a pool of $m$ candidate logistic predictors.
>
> Two "specialist" models are trained exclusively on one regime each (positive or negative inputs), achieving good performance on their respective regime and failing on the other.  The remaining $m-3$ "generalist'' predictors are trained on mixtures of the two regimes and therefore achieve inferior accuracy on each regime. The range of $p$ vs $(1-p)$ mixtures determines both the number of predictors, and their redundancy: predictors trained on close $p$ values are very similar.
>
> Additionally, we train one neural-network predictor, trained on combined data. It achieves better accuracy on each regime than any globalist, but inferior to specialists. Overall on average it is the best predictor.
> For any input $x$, **the optimal ensemble behavior is to select the specialist corresponding to the sign of $x$, and never to select one of the suboptimal generalists or the overall-best predictor**.
>
> We set our baseline experiment parameters as $d=100, k=30, \rho=0.0, m=11$.
> We then varied each. The results show that **across all settings IABMA yields better performance, and consistently prefers the correct specialist with much higher proportion.**
>
>
> **Base parameters:**
>
> | Method       | Accuracy         | ECE              | Correct specialist   | Best global   | Any generalist       |
> |:-------------|:-----------------|:-----------------|:---------------------|:-------------------|:-----------------|
> | MoE          | 0.904 (0.002) | 0.076 (0.003) | 0.000 (0.000)     | 1.000 (0.000)   | 0.000 (0.000) |
> | DLA          | 0.825 (0.007) | 0.100 (0.002) | 0.008 (0.001)     | 0.966 (0.005)   | 0.015 (0.003) |
> | SMC          | 0.680 (0.132) | 0.302 (0.155) | 0.051 (0.042)     | 0.481 (0.424)   | 0.414 (0.339) |
> | BHS          | 0.821 (0.007) | 0.097 (0.002) | 0.037 (0.034)     | 0.164 (0.018)   | 0.724 (0.036) |
> | DPP          | 0.910 (0.008) | 0.061 (0.017) | 0.227 (0.280)     | 0.773 (0.280)   | 0.000 (0.000) |
> | IABMA        | 0.919 (0.004) | 0.026 (0.007) | 0.948 (0.017)     | 0.037 (0.006)   | 0.015 (0.019) |
>
>
> **Higher number of overall features $d=300$:**
>
> | Method       | Accuracy         | ECE              | Correct specialist   | Best global   | Any generalist       |
> |:-------------|:-----------------|:-----------------|:---------------------|:-------------------|:-----------------|
> | MoE          | 0.858 (0.003) | 0.117 (0.003) | 0.000 (0.000)     | 1.000 (0.000)   | 0.000 (0.000) |
> | DLA          | 0.809 (0.005) | 0.068 (0.004) | 0.018 (0.005)     | 0.929 (0.004)   | 0.037 (0.001) |
> | SMC          | 0.816 (0.005) | 0.073 (0.004) | 0.000 (0.000)     | 1.000 (0.000)   | 0.000 (0.000) |
> | BHS          | 0.804 (0.005) | 0.067 (0.004) | 0.020 (0.002)     | 0.150 (0.005)   | 0.633 (0.011) |
> | DPP          | 0.858 (0.003) | 0.117 (0.003) | 0.000 (0.000)     | 1.000 (0.000)   | 0.000 (0.000) |
> | IABMA        | 0.882 (0.004) | 0.035 (0.001) | 0.552 (0.007)     | 0.163 (0.007)   | 0.285 (0.007) |
>
>
> **Higher number of informative features $k=50$:**
>
> | Method       | Accuracy         | ECE              | Correct specialist   | Best global   | Any generalist       |
> |:-------------|:-----------------|:-----------------|:---------------------|:-------------------|:-----------------|
> | MoE          | 0.882 (0.008) | 0.092 (0.008) | 0.000 (0.000)     | 1.000 (0.000)   | 0.000 (0.000) |
> | DLA          | 0.805 (0.009) | 0.087 (0.005) | 0.013 (0.003)     | 0.960 (0.008)   | 0.017 (0.003) |
> | SMC          | 0.813 (0.009) | 0.093 (0.008) | 0.000 (0.000)     | 1.000 (0.000)   | 0.000 (0.000) |
> | BHS          | 0.802 (0.008) | 0.084 (0.004) | 0.022 (0.002)     | 0.036 (0.004)   | 0.735 (0.013) |
> | DPP          | 0.882 (0.008) | 0.092 (0.008) | 0.000 (0.000)     | 1.000 (0.000)   | 0.000 (0.000) |
> | IABMA        | 0.902 (0.007) | 0.029 (0.006) | 0.675 (0.010)     | 0.325 (0.010)   | 0.000 (0.000) |

---

> ### Author Response · Authors · 2025-11-21
> **Scale Analysis - Part 2**
>
> **Higher number of predictors: $m=100$:**
>
> | Method       | Accuracy         | ECE              | Correct specialist   | Best global   | Any generalist       |
> |:-------------|:-----------------|:-----------------|:---------------------|:-------------------|:-----------------|
> | MoE          | 0.926 (0.005) | 0.038 (0.002) | 0.493 (0.009)     | 0.507 (0.009)   | 0.000 (0.000) |
> | DLA          | 0.758 (0.010) | 0.027 (0.005) | 0.006 (0.001)     | 0.955 (0.005)   | 0.035 (0.005) |
> | SMC          | 0.725 (0.067) | 0.104 (0.151) | 0.002 (0.004)     | 0.755 (0.355)   | 0.240 (0.347) |
> | BHS          | 0.757 (0.010) | 0.027 (0.005) | 0.114 (0.005)     | 0.019 (0.002)   | 0.867 (0.005) |
> | DPP          | 0.900 (0.002) | 0.081 (0.004) | 0.000 (0.000)     | 1.000 (0.000)   | 0.000 (0.000) |
> | IABMA        | 0.916 (0.004) | 0.028 (0.005) | 0.857 (0.008)     | 0.091 (0.007)   | 0.045 (0.002) |
>
> **More overlap between predictors $\rho=0.5$:**
>
> | Method       | Accuracy         | ECE              | Correct specialist   | Best global   | Any generalist       |
> |:-------------|:-----------------|:-----------------|:---------------------|:-------------------|:-----------------|
> | MoE          | 0.902 (0.006) | 0.073 (0.006) | 0.000 (0.000)     | 1.000 (0.000)   | 0.000 (0.000) |
> | DLA          | 0.797 (0.008) | 0.137 (0.002) | 0.001 (0.001)     | 0.996 (0.001)   | 0.002 (0.000) |
> | SMC          | 0.837 (0.008) | 0.163 (0.004) | 0.000 (0.000)     | 1.000 (0.000)   | 0.000 (0.000) |
> | BHS          | 0.787 (0.008) | 0.126 (0.006) | 0.022 (0.003)     | 0.019 (0.004)   | 0.952 (0.003) |
> | DPP          | 0.902 (0.006) | 0.073 (0.006) | 0.000 (0.000)     | 1.000 (0.000)   | 0.000 (0.000) |
> | IABMA        | 0.919 (0.006) | 0.022 (0.005) | 0.922 (0.001)     | 0.077 (0.001)   | 0.000 (0.000) |

---

> > ### Author Response · Authors · 2025-11-21
> > **Response -- part 2**
> >
> > Regarding your other comments:
> >
> > - In practice, formatting the results in figures 2 and 3 as tables takes a comparable amount of space, and we preferred a more visual representation that also shows the data construction in Fig. 2. We will try to include parts of the qualitative analysis from Appendix B.1 in the main text as you suggested.
> >
> > - Computational costs were reported in Appendix D.2. They show that the total runtime is comparable to MoE with the same network architecture.

---

> > > ### Comment · Reviewer_E3vv · 2025-11-25
> > >
> > > I thank the authors for their detailed response and the new experiments. However, after some thought and reading the other reviewer's comments I have decided to stay with my initial scoring.

---

> > > > ### Author Response · Authors · 2025-11-25
> > > >
> > > > Dear Reviewer,
> > > >
> > > > We appreciate the time you invested in reviewing our paper and understand the effort involved, as we also serve in various roles within the peer-review process.
> > > >
> > > > We respectfully note, however, that the purpose of the discussion period in a proper academic review process is to allow authors to address specific concerns raised by reviewers. If any concrete concerns remain, we would appreciate it if you could specify them.

---

### Official Review · Reviewer_2n4B · 2025-10-31

**Soundness:** 3
**Presentation:** 1
**Contribution:** 2
**Rating:** 2
**Confidence:** 4

**Summary:**

The authors study the interesting and important question of how to adaptively combine multiple prediction models, where here “adaptively” refers to a weighted mixture of the models with weights that depend on the test-point input covariates. The goal is both to improve overall performance relative to an individual model, as well as to improve “personalization,” meaning that the best model is used for a given input. The main claimed contribution is to present a *Bayesian* approach to adaptive model averaging--to contrast their contribution from prior work, the authors claim that “Previous adaptive approaches (see Section 1.1) addressed the task of specifying the adaptive weights $\alpha_j(x)$ from a *frequentist* point of view[...]” (emphasis added). The authors claim a theoretical guarantee that compares the proposed approach to the performance of the individual models, and they present experiments on simulated data, as well as on cancer drug-response prediction and credit-card fraud detection.

**Strengths:**

As mentioned, the authors study an important problem of how to best personalized combined models, that is, by taking an input-adaptive approach to model averaging. The approach seems reasonable, well-motivated, and empirical results appear okay. The theoretical analysis seems okay too, although I didn’t yet check it carefully due to the main weakness (see next section), for which is the main factor influencing my evaluation.

**Weaknesses:**

**Originality/significance:** In my view, the main weakness of the paper is that it does not make clear if/how the proposed methods differ from or improve on prior approaches to input-adaptive model averaging, including Bayesian approaches. In particular, there are at least two large categories of methods that I think at least the related work, and probably also the experiments section, should compare against more thoroughly: (1) mixture-of-experts models and (2) dependent Bayesian mixture models (eg, dependent Dirichlet mixture models or even input-dependent Gaussian mixture models).

For example, in the related work, the authors claim “Few methods assign input-dependent weights” before only providing one citation on mixture-of-experts, despite this area having a very large literature on data-dependent model averaging. It’s also not clear to me why MoE is not considered Bayesian. Eg, the authors could consult the following review papers:
- Masoudnia, S., & Ebrahimpour, R. (2014). Mixture of experts: a literature survey. Artificial Intelligence Review, 42(2), 275-293.
- Mu, S., & Lin, S. (2025). A comprehensive survey of mixture-of-experts: Algorithms, theory, and applications. arXiv preprint arXiv:2503.07137.
- Yuksel, S. E., Wilson, J. N., & Gader, P. D. (2012). Twenty years of mixture of experts. IEEE transactions on neural networks and learning systems, 23(8), 1177-1193.

Regarding dependent Bayesian mixture models, the authors could consult the following reviews, in particular with an eye to methods that are “covariate-dependent” mixture models:
- Barcella, W., De Iorio, M., & Baio, G. (2017). A comparative review of variable selection techniques for covariate dependent Dirichlet process mixture models. Canadian Journal of Statistics, 45(3), 254-273.
- Quintana, F. A., Müller, P., Jara, A., & MacEachern, S. N. (2022). The dependent Dirichlet process and related models. Statistical Science, 37(1), 24-41.

**Questions:**

Could the authors please clarify how the proposed methods relate to existing literature on mixture of experts and/or input/covariate-dependent Bayesian mixture models? This seems necessary to properly understanding the paper’s contribution, and I didn’t find this sufficiently discussed in the paper.

---

> ### Author Response · Authors · 2025-11-18
> **Response**
>
> Dear reviewer,
>
> Thank you for the provided references and feedback.
>
> Our setting fundamentally differs from both the mixture-of-experts (MoE) and dependent Dirichlet Processes (DDP). Our method assumes a **fixed set of pre-trained predictors**, whereas both classical MoE and DDP approaches **jointly train the experts and the weights**.
>
> Thus, vast literature explores combinations of expert classes (e.g., linear models, MLPs, deep CNNs) and gating architectures. However, among these methods, the only comparable variant to ours is an MoE applied **on top of pre-trained experts**, which is the comparison we included (with the same gating architecture as our model).
>
> Placing a prior on the network parameters in MoE acts as regularization on the gate; it does not define a prior on the mixture weights themselves. In contrast, IABMA introduces a prior directly over the mixture-weights, which are then inferred as posterior probabilities $p(g(x)=e_j \mid x, \mathcal{D}). This yields a probabilistic model where weights represent estimated posterior probabilities rather than optimized scores.
>
> As for dependent Dirichlet processes (DDP), these models typically define a nonparametric prior over infinitely many latent components (and jointly learn both component parameters and covariate-dependent weights). Nevertheless, to address your suggestion, we added a comparison to an **adapted DDP model**: we fix the “atoms” to be the pre-trained experts, and learn covariate-dependent “stick-breaking” weights over them. The results below (accuracy for classification, RMSE for regression) show that, in all cases IABMA performs better than the adapted DPP, in some cases with large improvements.
>
> | experiment                        | IABMA | DDP   | % change |
> |-----------------------------------|-------|-------|----------|
> | Fraud (Accuracy ↑)                | 0.736 | 0.695 | +5.90%    |
> | Cancer (RMSE ↓)                   | 1.842 | 1.863 | −1.13%    |
> | Synthetic_binary (Accuracy ↑)     | 0.813 | 0.804 | +1.12%    |
> | Credit-g (RMSE ↓)                 | 0.676 | 0.655 | +3.21%    |
> | Spambase (RMSE ↓)                 | 0.764 | 0.754 | +1.33%    |
> | Bike-sharing (Accuracy ↑)     | 0.446 | 0.479 | −6.89%    |
> |California-housing (RMSE ↓)   | 0.024 | 0.031 | −22.58%   |
>
> Like MoE, DLA, SMC, BHS, in DDP  adaptivity is achieved through weights that vary with $x$. IABMA adds a second layer of adaptivity: **the prior itself varies with $x$**, linking each expert’s prior selection probability to its **expected likelihood** and propagating this uncertainty into the posterior.
>
> In the revised version, we will explicitly highlight both the applicability to pre-trained models and the additional layer of adaptivity arising from the input-dependent prior.

---

> > ### Comment · Reviewer_2n4B · 2025-11-25
> >
> > I thank the authors for their response, in particular for their explanation of how their method compares to MoE and DDPs and for the added experimental comparisons. I understand that the authors claim that their method is in a different setting than MoE and DDPs (ie, the authors focus on averaging "a fixed set of pre-trained predictors, whereas both classical MoE and DDP approaches jointly train the experts and the weights"). However, with this explanation, I still think that the paper could be better contextualized within related work, for instance discussing how the paper relates to the large literature on Bayesian model averaging, and learning to defer to multiple experts (where in my understanding the MoE is not always jointly trained).
> >
> > I have a follow-up question about the new experiment provided above. In their rebuttal, the authors state "in all cases IABMA performs better than the adapted DDP, in some cases with large improvements," but I only count 4/7 cases where the reported numbers are better for the proposed IABMA method than for the DDP baseline. That is, in Credit-g and Spambase it seems that RMSE *increases* and for bike sharing accuracy *decreases*. Can the authors please clarify if the statement or the results were incorrect?

---

> > > ### Author Response · Authors · 2025-11-25
> > > **Response**
> > >
> > > Dear reviewer,
> > >
> > > We will expand the related work to include additional discussion on Bayesian model averaging as well as relevant MoE variants. We will upload soon a revised version.
> > >
> > > Regarding your question -- sorry for the confusion, we were referring to the main experiments, which are the heteroscedastic settings: Fraud, Cancer, and the Synthetic benchmarks. The substantial change appears in the Fraud experiment. The UCI datasets, as noted, exhibit low heteroscedasticity, where no method is expected to consistently outperform others.
> > >
> > > While the DPP baseline is not an established method but rather a new baseline we introduced in response to your suggestion, we would like to ask you to look at the additional experimental results we posted in response to Reviewer E3vv, where we included the DPP baseline as well. These as well will be included in the revised version.

---

### Official Review · Reviewer_ozkA · 2025-11-01

**Soundness:** 2
**Presentation:** 2
**Contribution:** 2
**Rating:** 2
**Confidence:** 5

**Summary:**

This paper proposes a framework for combining multiple predictive models by assigning input-specific weights in a Bayesian manner. Unlike classical Bayesian Model Averaging (BMA), which uses global model weights, IABMA introduces an input-dependent prior over model-selection functions, leading to input-adaptive posterior weights. The posterior is approximated using amortized variational inference, yielding instance-specific model weightings. The authors derive a finite-sample likelihood guarantee showing that the proposed predictor performs competitively with the best per-input model selector.

**Strengths:**

Conceptual coherence: The probabilistic formulation that derives adaptive weights from a Bayesian posterior is principled and internally consistent.

Empirical breadth: The experiments span both regression and classification, synthetic and real datasets, with comparisons to multiple baselines.

**Weaknesses:**

1.	Limited novelty: The proposed approach is largely a Bayesian reinterpretation of existing adaptive ensemble methods such as Mixture of Experts and Bayesian Hierarchical Stacking . The key innovation, introducing an input-dependent prior, is conceptually modest and primarily repackages known ideas in new notation.

2.	Superficial theoretical development: The likelihood guarantee is a straightforward adaptation of Jensen’s inequality, offering minimal insight into the behavior of the amortized inference procedure or its generalization properties. No convergence analysis, uncertainty quantification, or theoretical justification for the variational approximation is provided.

3.	Lack of methodological clarity: The construction of the input-adaptive prior is heuristic, based on an “energy” integral that lacks intuitive interpretation and appears computationally impractical for high-dimensional continuous outcomes. The method relies on ad-hoc Monte Carlo approximations, raising concerns about scalability and stability.

4.	Amortized inference design is under-specified: The paper treats the amortized posterior network as a black box, without ablation or sensitivity analysis on architecture, optimization, or overfitting. It is unclear whether performance gains come from the variational parameterization or the adaptive prior itself.

5.	Empirical evidence is weakly convincing: Improvements over baselines are small and inconsistent across datasets; Comparisons do not include recent or stronger baselines in adaptive ensembling (e.g., deep mixture-of-experts architectures); Some tasks (e.g., PRISM, fraud detection) lack details on train/test splits, data leakage control, and statistical significance of reported differences.

6.	Overstated claims: The paper claims “formal guarantees” and “Bayes-optimal adaptive weights,” but these rely on unverified approximations and assumptions. The results fall short of demonstrating real-world robustness or interpretability advantages.

7.	Expository issues: While the paper is long and dense, it lacks intuition; the heavy notation and abstract measure-theoretic framing (e.g., pushforward arguments) obscure rather than clarify the contribution.

**Questions:**

How does IABMA differ in substance from Mixture of Experts with a Bayesian treatment of gating? What new insights or properties does the input-dependent prior confer?

How sensitive is the model to the design of the energy-based prior and the range of integration for continuous outcomes?

What guarantees (if any) can be provided for the variational approximation quality or convergence of amortized inference?

Could the same adaptive weighting effect be achieved more simply with a discriminatively trained gating network, without the Bayesian formalism?

What is the computational complexity of evaluating Eq. (9) and optimizing the ELBO in large-scale settings?

---

> ### Author Response · Authors · 2025-11-14
> **Response -- part 1**
>
> Dear reviewer,
>
> Thank you for reviewing our paper. Please see below.
>
> Weaknesses:
>
> 1. The novelty of our approach consists of two main components:
>
> A. Our Bayesian formulation defines what the adaptive weights should be. Each pair $(x,y)$ is assumed to be generated by one of the candidate predictors $f_j$, and the adaptive weights naturally arise as the posterior probabilities $p(g(x)=e_j \mid x, \mathcal{D})$ that predictor $j$ generated the pair.
> In contrast, mixture-of-experts (MoE) models assign weights via maximum likelihood without a generative model. As a result, in training their gating networks learns to favor the predictor most confident about the observed outcome $y$, regardless of the true confidence.
>
> Our approach is therefore not a reinterpretation of MoE
> highlight this distinction: our posterior-based weights adapt to the predictor that best explains $(x,y)$, whereas MoE consistently favors the dominant experts even where weaker predictors are more appropriate.
>
> B. Since the true posterior probabilities are unknown, we propose a method to estimate them. Our approach specifies an adaptive prior, and performs amortized inference of the posterior.
> The adaptive prior links each model’s selection probability for a new input $x$ (with unknown $y$) to its expected likelihood over possible outcomes $y$, thereby propagating predictive uncertainty into the posterior estimation.
> Unlike a classical posterior that conditions on observed data, this posterior is contextualized by a specific test input $x$. To capture this dependence, we employ an amortized inference network that takes $x$ as input and outputs an approximate posterior distribution tailored to that input.
>
> 2. While the likelihood guarantee is simple to prove, it offers a nontrivial insight: the posterior adaptive mixture is never worse (in expectation) than any individual predictor, up to an additive penalty reflecting each expert’s average posterior weight. That means it can choose a different expert $f_j$ for each input $x$, and still the posterior averaging can not be much worse.
> This extends the classical result that global averaging outperforms fixed model selection.
> Regarding uncertainty quantification, the true posterior provides exact predictive uncertainty:
> $p(y\mid x,\mathcal{D})=\sum_{j=1}^{m}f_{j}(y\mid x)\,p(g(x) = e_j \mid x,\mathcal{D})$ (equation 3).
> Any deviation arises solely from approximating the posterior. minimizes the KL divergence between the true and amortized posteriors via an ELBO objective, so the approximation error is precisely quantified by this KL term, which depends on the richness of the specified variational family.
>
> 3. The energy prior is the marginalization over possible outcomes $y$, linking each model’s expected likelihood to its selection probability for a given input $x$, instead of a fixed prior. For discrete $y$ this reduces to an explicitly computed sum. For continuous $y$ the integral is approximated by Monte Carlo sampling. We observed no stability issues in any experiments, and the achieved performance gains were consistent across both classification (discrete $y$) and regression (continuous $y$) tasks.
>
> 4. The prior and the posterior are part of the same probabilistic model and thus not meant to be applied separately: since the prior over selectors $g$ depends on the input $x$, adaptivity in our model arises jointly from both the variability of $g(x)$ and the input-dependent prior.
>
> Nevertheless, we performed experiments with a uniform prior over selectors. The results below (accuracy for classifications, RMSE for regression) show that without the adaptive prior,  our method still outperforms most baselines but with a smaller margin. We will add those in the supplementary material of the revised version.
>
> | experiment             | Best-single | Uniform Avg | Freq Avg | BMA    | MoE    | DLA    | SMC    | BHS    | IABMA  | IABMA ablation |
> |------------------------|-------------|-------------|----------|--------|--------|--------|--------|--------|--------|----------------|
> | synthetic_binary  | 0.797       | 0.759       | 0.774    | 0.790 | 0.807 | 0.798 | 0.807 | 0.798 | 0.813 | 0.813          |
> | fraud             | 0.657       | 0.670       | 0.669    | 0.667 | 0.713 | 0.689 | 0.653 | 0.670 | 0.736 | 0.713          |
> | cancer                 | 1.927       | 1.870       | 1.853    | 1.860  | 1.856  | 1.861  | 1.897  | 1.870  | 1.842  | 1.853          |
> | uci-bike-sharing       | 0.582       | 0.491       | 0.448    | 0.447  | 0.581  | 0.433  | 0.483  | 0.491  | 0.446  |  0.451          |
> | uci-california-housing | 0.036       | 0.025       | 0.025    | 0.029  | 0.035  | 0.025  | 0.029  | 0.025  | 0.024  | 0.024          |
> | uci-credit-g      | 0.634       | 0.684       | 0.662    | 0.648 | 0.624 | 0.668 | 0.626 | 0.682 | 0.676 | 0.575          |
> | uci-spambase      | 0.699       | 0.702       | 0.738    | 0.760 | 0.760 | 0.729 | 0.757 | 0.646 | 0.764 | 0.739          |

---

> ### Author Response · Authors · 2025-11-14
> **Response -- part 2**
>
> 5. We ensured fair comparison by using the same baseline network architecture for both our method and the mixture-of-experts (MoE) model across all experiments. All reported results include standard deviations, and we have now added formal statistical tests (see below).
> The advantages of adaptive model averaging appear most clearly in **heteroskedastic settings**. In all three such experiments—synthetic, PRISM cancer drug-response, and fraud detection—our method **statistically significantly** outperforms all competitors. Reported below are one-sided t-test results with Benjamini–Hochberg correction for multiple comparisons (accuracy for classification, RMSE for regression). Improvements in calibration (ECE) are also statistically significant (see Figures 2–3).
>
> | experiment       | BMA       | MoE       | DLA       | SMC       | BHS       |
> |:-----------------|:----------|:----------|:----------|:----------|:----------|
> | cancer           | <0.01 | <0.01 | <0.01 | <0.01 | <0.01 |
> | fraud            | <0.01 | <0.01 | <0.01 | <0.01 | <0.01 |
> | synthetic_binary | <0.01 | 0.054 | <0.01  | 0.037 | <0.01 |
>
> The UCI datasets serve as low-heteroskedastic benchmarks, where large consistent gains are not expected. Even in these settings, our method improves RMSE in both regression tasks and accuracy in both classification tasks (Credit-G, Spambase). Roughly half of these improvements are statistically significant (9 of 20 pairwise comparisons) ( over 5 competitors across 4 UCI datasets).
>
> | experiment             | BMA       | MoE       | DLA       | SMC       | BHS       |
> |:-----------------------|:----------|:----------|:----------|:----------|:----------|
> | uci-credit-g     | 0.104     | <0.01 | 0.338     | <0.01 | 0.758     |
> | uci-spambase     | 0.349     | 0.380     | 0.019 | 0.349     | 0.016 |
> | uci-bike-sharing       | 0.316     | <0.01 | 1.000     | <0.01 | <0.01 |
> | uci-california-housing | <0.01 | <0.01 | 0.153     | 0.153     | 0.153     |
>
> 6. The guarantees and the Bayes-optimal adaptive weights refer to the ideal Bayesian formulation under the stated model assumptions. The practical implementation uses variational approximation to estimate these quantities. We can emphasize this more.
>
> 7.  Formal notation (e.g., pushforward measures) is necessary to make the Bayesian construction precise, as the relevant distributions are not defined on the same space otherwise. To improve clarity, we have minimized such notation in the main text—now stating only that “a formal proof of this equality is outlined in Appendix A.1”.
>
> **Questions:**
>
> 1. In a Bayesian treatment of Mixture-of-Experts (MoE), priors can be placed on the gating network’s parameters or the gating probabilities themselves, but the gating probabilities do not correspond to any posterior distribution. They are optimized to fit $y$, not inferred as posteriors conditioned on training $x$ and $y$ values, and test $x$.
> In contrast, IABMA defines a generative model where each pair $(x,y)$ is assumed to be produced by one expert. The adaptive weights then arise as posterior probabilities representing how likely each expert is to have generated the observation. Thus, IABMA infers weights from the Bayesian model itself.
>
> 2. The input-dependent prior gives the model a way to anticipate which experts are likely to perform well before observing $y$. It **links each expert’s prior selection probability for a new input $x$ to its expected likelihood over possible outcomes.**
> This regularizes the posterior toward experts with higher expected predictive accuracy at each $x$.
>
> 3.  We addressed the sensitivity of the model to the design of the energy-based prior and the range of integration for continuous outcomes in weakness 3. Please see above.
>
> 4. As in standard variational inference, the amortized posterior is obtained by optimizing the ELBO, which is equivalent to minimizing the KL divergence between the approximate and true posterior. The quality of the approximation is therefore exactly quantified by this KL term, which is determined by the expressiveness of the chosen variational family (including the inference network).
>
> 5. Without the Bayesian formalism, a discriminatively trained gating network can learn adaptive weights, but not the same ones. The gating scores are learned functions optimized for predictive likelihood, but with no equivalent parts to (i) a prior expressing which expert is expected to generate a given input, and (ii) a posterior that accounts for uncertainty conditioned on both the new input $x$ and the training data (x,y).
>
> 6. In classification tasks with discrete $y$ Eq. (9) is a simple summation. For continuous $y$, we use 64 Monte Carlo samples in all experiments. As shown in Table 8, total runtime remains comparable to Mixture-of-Experts with the same network architecture.

---

> > ### Comment · Reviewer_ozkA · 2025-11-26
> >
> > I thank the authors for the detailed response to my questions. I will raise my score.

---

> > > ### Author Response · Authors · 2025-11-26
> > > **Response**
> > >
> > > Thank you.
> > > We would appreciate it if you could specify any concrete remaining concerns if there are any.

---

### Official Review · Reviewer_nu4b · 2025-11-06

**Soundness:** 3
**Presentation:** 3
**Contribution:** 3
**Rating:** 6
**Confidence:** 1

**Summary:**

The paper studies a Bayesian way to combine different models.

**Strengths:**

Cannot assess

**Weaknesses:**

cannot assess

**Questions:**

None

---

### Author Response · Authors · 2025-11-28
**Rebuttal summary**

Dear reviewers and AC,

We uploaded a revised version that includes:
- Additional baseline comparison with a DDP model, adapted to operate on pre-trained predictors.
- Scale and sensitivity study consisting of 5 additional experiments.
- Formal statistical significance tests supporting reported improvements.
- Ablation showing the contribution of the input-adaptive prior.

We have also expanded discussion of all points raised in the reviews, including:
- The novelty of our approach and its Bayesian formulation, and the fact that the posterior predictive distribution yields calibrated uncertainty and is fully recoverable within our model (as the posterior is set to the true family).
- The implications of assuming pre-trained predictors, including why only certain variants of MoE and DDP are applicable, and why the internal complexity or capacity of the predictors becomes immaterial.
- The role of the UCI datasets as low-heteroskedastic benchmarks, used to verify that IABMA does not degrade performance when adaptivity is not necessary.

---

### Meta-Review · Area_Chair_Ky5b · 2026-01-06

**Summary:**

In this paper, the authors propose a novel input-dependent Bayesian model averaging approach for combining multiple predictive models. The reviewers acknowledge the significance of the problem and note that the paper is generally coherent, with the main ideas easy to follow. However, they raised serious concerns regarding: (a) the relationship of the proposed work to state-of-the-art methods, including mixture-of-experts and dependent Bayesian approaches, and (b) insufficient experimental validation. In the rebuttal, the authors provided responses that partially clarified some points, but key issues remain unresolved. I believe the paper requires substantial revision to position it more clearly against existing  approaches and to address all reviewers’ concerns. Therefore, I recommend rejecting the paper at this stage and encourage the authors to incorporate the feedback, strengthen the work, and consider resubmission to a future venue.

**Reviewer Concerns:**

Reviewers ozkA and 2n4B raised concerns about the originality and significance of the proposed approach compared to prior work, such as mixture-of-experts. The authors partially addressed these concerns by explaining that, unlike MoE methods, their approach uses fixed pretrained predictors and learns only adaptive weights rather than jointly learning both components. While this argument was partly accepted by reviewers, I believe that further clarification is needed. Making clear why this key point of the proposed approach is benificial over other approaches is critical. Specifically, although using pretrained encoders reduces computational demands, it is unclear why this should lead to superior performance over approaches that learn both weights and predictors. Additionally, reviewers ozkA and E3vv expressed concerns about insufficient empirical validation. The authors provided additional experimental results during the rebuttal, which improved the empirical evidence, but questions remain, for example, it is still unclear why IABMA outperforms dependent DPP.

**Reviewer Scores:**

Reviewer ozkA was satisfied with the authors' responses and would probably have raised the score, while  E3vv was not fully convinced and said that they would keep their initial score. Reviewer 2n4B still had same questions after initial rebuttal and it's not quite clear if final answers provided by the authors would convince them to raise their score.

---

### Decision · Program_Chairs · 2026-01-26

Reject